# Distinct responses of Purkinje neurons and roles of simple spikes during associative motor learning in larval zebrafish

**Thomas C Harmon[1,2], Uri Magaram[1], David L McLean[1,2], Indira M Raman[1,2]***

[1]Department of Neurobiology, Northwestern University, Evanston, United States;
[2]Interdepartmental Neuroscience Program, Northwestern University, Evanston, United States

**Abstract** To study cerebellar activity during learning, we made whole-cell recordings from larval zebrafish Purkinje cells while monitoring fictive swimming during associative conditioning. Fish learned to swim in response to visual stimulation preceding tactile stimulation of the tail. Learning was abolished by cerebellar ablation. All Purkinje cells showed task-related activity. Based on how many complex spikes emerged during learned swimming, they were classified as multiple, single, or zero complex spike (MCS, SCS, ZCS) cells. With learning, MCS and ZCS cells developed increased climbing fiber (MCS) or parallel fiber (ZCS) input during visual stimulation; SCS cells fired complex spikes associated with learned swimming episodes. The categories correlated with location. Optogenetically suppressing simple spikes only during visual stimulation demonstrated that simple spikes are required for acquisition and early stages of expression of learned responses, but not their maintenance, consistent with a transient, instructive role for simple spikes during cerebellar learning in larval zebrafish.

*For correspondence: i-raman@northwestern.edu

## Introduction

The activity of cerebellar Purkinje cells regulates both practiced and new movements (*Thach, 1968*; *McCormick and Thompson, 1984*; *Medina et al., 2000*; *Mauk et al., 2014*). In vertebrates from fish to mammals, Purkinje cells influence motor behavior via both simple and complex spikes (*Eccles et al., 1966*; *Monsivais et al., 2005*; *Khaliq and Raman, 2005*; *Han and Bell, 2003*). Simple spikes occur spontaneously and are modulated by synaptic input from granule cells and inhibitory interneurons; the resulting activity alters firing patterns of Purkinje target neurons, whose output generates movements (*Thach, 1968*; *McDevitt et al., 1987*; *Witter et al., 2013*; *Heiney et al., 2014*; *Lee et al., 2015*). Complex spikes arise from synaptic input from climbing fibers and can induce plasticity of other afferents to Purkinje cells, thereby serving as teaching and/or error signals during motor learning (*Gilbert and Thach, 1977*; *Mauk et al., 1986*; *Medina et al., 2002*; *Ohmae and Medina, 2015*).

Despite their shared modes of action potential firing, Purkinje cells in different cerebellar regions contribute differentially to behaviors, owing to their anatomical connections, including distinct innervation patterns by mossy fiber-granule cell pathways (*Bower et al., 1981*; *Bower and Woolston, 1983*; *Garwicz et al., 1998*) and inferior olivary modules (*Voogd and Glickstein, 1998*; *Sugihara and Shinoda, 2004*; *Ruigrok, 2011*; *Cerminara and Apps, 2011*), as well as different targets. Among Purkinje cells engaged by a particular action, learning often correlates with the emergence of new patterns of activity. In primates, ferrets, rabbits, and mice, the rate and timing of

simple spikes and/or complex spikes can change as animals acquire novel motor behaviors (*Gilbert and Thach, 1977*; *Jirenhed et al., 2007*; *Halverson et al., 2015*; *Ohmae and Medina, 2015*; *ten Brinke et al., 2015*). In vitro studies have demonstrated many forms of synaptic plasticity that may underlie these changes (*Hansel et al., 2001*; *Ito et al., 2014*). Few in vivo preparations are available, however, in which synaptic changes and the resultant spikes can be monitored and manipulated over the full time course of learning.

Here, we explored whether larval zebrafish might offer such a preparation. At 6–8 days post-fertilization (dpf), the zebrafish cerebellum has a relatively simple structure, containing ~300 Purkinje cells (*Hamling et al., 2015*). Zebrafish larvae swim and learn to alter their movements in response to sensory stimulation (*Portugues and Engert, 2011*; *Mu et al., 2012*; *Amo et al., 2014*; *Matsui et al., 2014*; *Pantoja et al., 2016*), and they display some forms of motor learning that depend on the cerebellum (*Aizenberg and Schuman, 2011*; *Ahrens et al., 2012*). Additionally, larval zebrafish Purkinje cells generate both simple and complex spikes (*Hsieh et al., 2014*; *Sengupta and Thirumalai, 2015*), providing a potentially useful system to examine how these conserved signals may contribute to cerebellar learning in different species (*Scalise et al., 2016*).

We therefore developed a cerebellar associative learning task for immobilized larval zebrafish, and made whole-cell recordings of Purkinje cell activity (1) in response to visual and tactile sensory stimuli, (2) during episodes of spontaneous, reflex, and learned swimming and (3) before, during, and after training. Three populations of Purkinje cells could be distinguished by their complex spike responses after conditioning, as well as by their topographical location in the cerebellum. Optogenetically suppressing simple spikes in all Purkinje cells during training further showed that the role of simple spikes changes as learned movements emerge, such that learning can be abolished and/or altered depending on when simple spikes are disrupted.

## Results

### Purkinje cell recordings

With the goal of testing how Purkinje neurons contribute to associative learning in larval zebrafish, we first investigated whether they displayed consistent synaptic responses and firing patterns during sensory stimuli and/or fictive swimming. In immobilized fish, whole cell recordings from Purkinje cells and extracellular recordings from ventral roots in the tail were made simultaneously (*Figure 1A and B*). Purkinje cells were located in the most superficial cell body layer of the corpus cerebelli (*Bae et al., 2009*) and had a single apical dendrite in the molecular layer (*Figure 1A and C*). Under voltage-clamp, spontaneous synaptic activity was evident as large-amplitude EPSCs ($-253 \pm 29$ pA, N = 39 cells), likely from climbing fibers, and small-amplitude EPSCs, generally <20 pA, likely from parallel fibers (*Figure 1D and E*). Current-clamp recordings from these cells revealed large-amplitude complex spikes and small-amplitude simple spikes (*Figure 1F*, *top 1G*), further confirming their identity as Purkinje cells (*Hsieh et al., 2014*; *Sengupta and Thirumalai, 2015*; *Scalise et al., 2016*). As expected, basal firing rates were lower for complex spikes than for simple spikes ($0.3 \pm 0.03$ *vs.* $6.4 \pm 1.2$ spikes/s, N = 42 cells; p<0.001, paired t-test). These values are in good agreement with previous studies in larval zebrafish (*Scalise et al., 2016*). Subthreshold EPSPs, likely arising from parallel fibers, were also evident (*Figure 1F*, *bottom*). In some records, increased parallel fiber activity correlated with episodes of fictive swimming that occurred spontaneously. In voltage clamp, this activity was evident as clusters of EPSCs (*Figure 1E and H*); in current clamp these events could summate to produce long-lasting depolarizations, typically with simple spikes riding on top (*Figure 1I*).

### Identification of events

In each cell, complex spikes were larger in amplitude and rose faster than simple spikes. Their absolute amplitudes and rise times varied from event to event, however, since variations in membrane potential (e.g., from summating EPSPs or hyperpolarization) could alter driving force on synaptic currents or inactivate/recover voltage-gated channels. Amplitudes and rise times were also influenced by the magnitude of the underlying synaptic conductances, which varied from cell to cell. Since subsequent analyses relied on distinguishing climbing-fiber-driven and parallel-fiber-driven events, we (1) identified complex spikes based on rate of rise, (2) confirmed by inspection that they were large

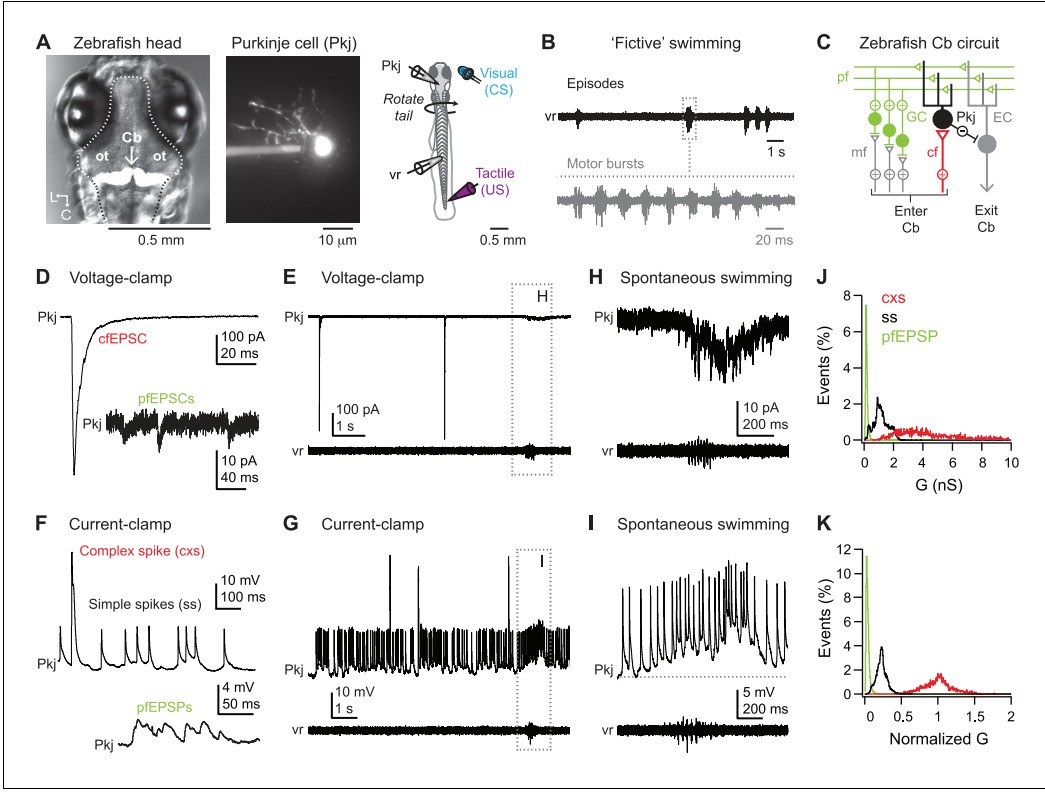

**Figure 1.** Spontaneous activity during dual recordings of Purkinje cells and ventral roots. (**A**) Left, dorsal view of larval zebrafish head illustrating the location of the cerebellum (Cb, arrow). In this image, cerebellar Purkinje cells are fluorescently labeled by Archaerhodopsin-3 (white). Dotted line, outline of brain. ot: optic tectum. L, lateral; C, caudal. Middle, a Purkinje cell filled with Alexa Fluor 488. Right, schematic of the preparation. (**B**) Sample ventral root (vr) recording showing episodic bouts of fictive swimming (compressed time base, black) and cyclical motor bursts (expanded time base, grey). (**C**) Schematic of the zebrafish cerebellum. Pkj: Purkinje cell. EC: eurydendroid cell; GC: granule cell; cf: climbing fiber; mf: mossy fibers; pf: parallel fibers; Cb: cerebellum. (**D**) Sample voltage-clamp recording (holding potential = −60 mV, all voltage-clamp records) of cfEPSC and pfEPSCs (inset). (**E**) Sample voltage-clamp recording (top) and simultaneous vr recording (bottom). Box, episode of spontaneous swimming, expanded in (**H**). (**F**) Sample current-clamp recording of complex and simple spikes (top) and pfEPSPs (bottom). (**G**) Sample current-clamp recording of Purkinje cell spikes and simultaneous vr recording. Box, episode of spontaneous swimming, expanded in (**I**). Recordings in (**E**) and (**G**) are from the same cell. (**H**) Higher gain voltage-clamp and vr recording from (**E**) of clustered parallel fiber EPSCs during spontaneous swimming. (**I**) Higher gain current-clamp and vr recording from (**G**) of a parallel fiber driven long-lasting depolarization and simple spikes during spontaneous swimming. Dotted line, inter-spike potential (−56 mV) to illustrate depolarization. (**J**) Distribution of conductances associated with all complex spikes (cxs), simple spikes (ss), and parallel fiber EPSPs (pfEPSPs) included in the study. Absolute conductances (bin width, cxs and ss = 0.02 nS; pfEPSP = 0.01 nS). (**K**) As in (**J**) but conductances normalized to the mean cxs conductance in each cell (bin width = 0.005).

events rising directly from the baseline, as expected for synaptic currents exceeding intrinsic currents, and (3) verified their identity as climbing-fiber-dependent by estimating the conductance underlying the upstroke of the event. With complex spikes extracted from the record, we identified simple spikes based on rate of rise, visually confirmed that they had an inflection on the upstroke, consistent with activation of voltage-gated channels, and estimated the underlying conductance. Lastly, with simple spikes extracted, we identified EPSPs based on rise rate, confirmed them by inspection, and calculated the conductance (see *Materials and methods*).

In the complete study, we identified 1930 complex spikes, 16,763 simple spikes, and 21,096 EPSPs in 55 cells. The mean conductance (± S.D.) associated with complex spikes was 4.86 ± 0.25 nS; with simple spikes, 1.17 ± 0.5 nS; and with EPSPs, 0.17 ± 0.08 nS. These values correspond to about −300 pA at −60 mV for the climbing fiber EPSC and −10 pA at −60 mV for parallel fiber

EPSCs, consistent with voltage-clamp recordings. They also predict about −100 pA of Na current around −30 mV on the upstroke of the simple spike. The complete distribution of all conductances in all cells (*Figure 1J*) showed that the overlap of the complex and simple spike distributions was 15.2%. This overlap does not reflect the likelihood of misidentification, however, because complex and simple spikes were distinguishable within each record, by the absence (complex spike) or presence (simple spike) of an inflection as well as by the conductance normalized to the mean complex spike-associated conductance within each cell. Plotting the conductances for simple spikes relative to those for complex spikes in the same cells gave an overlap of 0.66% with all cells pooled (*Figure 1K*). When these normalized measurements were made on a cell-by-cell basis, the overlap fell to 0.08 ± 0.06%. A subset of cells showed EPSPs but were too depolarized to fire simple spikes (*Materials and methods*); in these cells the conductance associated with the complex spike was ≥2 S.D. from the mean conductance of the simple spike distribution. Thus, simple and complex spikes could be distinguished with little error.

Similar analyses gave a simple spike-EPSP overlap of 9.8% for the non-normalized distribution across all cells (*Figure 1J*), 5.2% for the normalized distribution across all cells (*Figure 1K*), and 2.1 ± 0.6% within cells. EPSPs and simple spikes could thus be distinguished; however, our primary interest in these events was to identify parallel-fiber-dependent, synaptically driven spikes, rather than spontaneous spikes, during cerebellar learning. Therefore, we focused subsequent analyses on parallel-fiber EPSPs (pfEPSPs) that led to simple spikes rather than on simple spikes directly.

## Heterogeneity of Purkinje cell responses during motor behavior and sensory stimulation

To begin to examine task-related Purkinje cell activity, we first assessed responses during spontaneous motor behavior. During 39 of 49 Purkinje cell recordings, spontaneous fictive swimming occurred. All 39 Purkinje cells modulated their activity during spontaneous swimming, but not all cells responded in the same way. The variety of responses is catalogued here to provide the context for the studies of motor learning described below.

Many cells generated complex spikes during spontaneous swimming (N = 29/39; *Figure 2A*); in 11 of these cells, complex spikes occurred on every swimming episode, while the others produced complex spikes on 53.4 ± 4.8% of episodes. Most Purkinje cells also showed EPSPs with long-lasting depolarizations (>200 ms) that evoked simple spikes, which could outlast swimming (N = 26/39; *Figure 2A*, *top three panels*), while others showed long-lasting hyperpolarizations of 5 to 10 mV (N = 9/39 cells; *Figure 2A*, *bottom*). These observations are consistent with previous descriptions of spontaneous, motor-related Purkinje cell responses in larval zebrafish (*Sengupta and Thirumalai, 2015*).

Next, to examine Purkinje cell responses to sensory input, we presented fish with a high-contrast blue light (the 'visual' stimulus, 2 s) or a brief, mild electrical stimulus to the tip of the tail (the 'tactile' stimulus, 5 ms). With repeated presentations, each sensory stimulus evoked a consistent response in each Purkinje cell. As in the case of spontaneous swimming, however, responses varied across the population of Purkinje cells. With high-contrast visual stimuli, both parallel and climbing fiber responses were observed in 12 of 49 cells, while other cells showed only climbing fiber responses (N = 16/49), only parallel fiber responses (N = 10/49), or no detectable change in activity (N = 11/49; *Figure 2B*, *top to bottom*).

In 82% of recordings (N = 40/49), high-contrast visual stimulation also evoked fictive swimming (*Figure 2C*). In some fish, swimming occurred a few hundred ms after light onset, with or without another episode of swimming after light offset (N = 22/40); other fish swam only at the offset of the light (N = 18/40). The swimming latencies are consistent with the relatively long delays reported for visuomotor behavioral responses in zebrafish (*Brockerhoff et al., 1995*; *Wang and McLean, 2014*; *Portugues et al., 2015*).

With tactile stimuli, fictive swimming was evoked with a short latency (16.9 ± 1.3 ms after stimulation) and with 100% reliability (N = 49/49 fish; *Figure 2D*). This invariant response is consistent with the involvement of reflexive brainstem pathways responsible for evasive swimming maneuvers in larval zebrafish (*Bhatt et al., 2007*; *Lacoste et al., 2015*; *Koyama et al., 2016*), and validates the tactile stimulus for use as an unconditional stimulus in later associative learning experiments. Additionally, 86% of Purkinje cells responded to the first tactile stimulus with at least one complex spike (N = 42/49 cells).

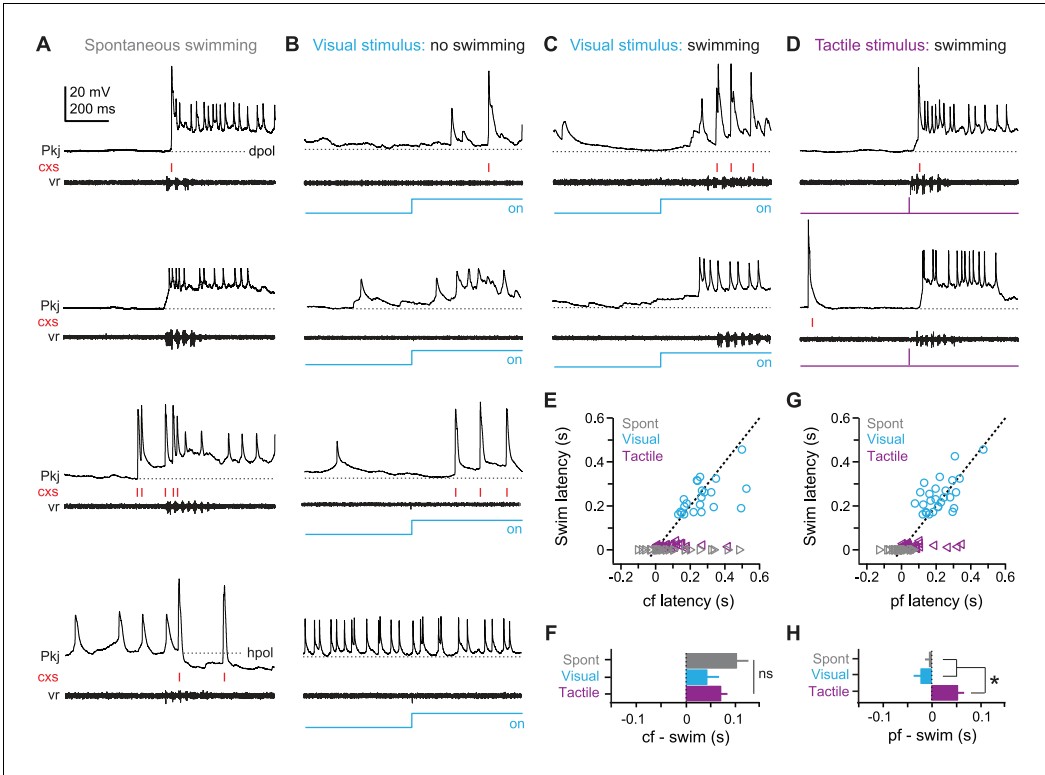

**Figure 2.** Purkinje cell responses during sensory stimuli and motor commands associated with fictive swimming.
(**A**) Responses of four different Purkinje cells (top to bottom) during spontaneous swimming, showing different combinations of climbing fiber, parallel fiber, and putative inhibitory input, resulting in complex spikes, simple spikes with long-lasting depolarizations, and/or hyperpolarization. In all panels, complex spikes (cxs) are indicated with red tick marks and the corresponding ventral root recording is included. Dotted lines at inter-spike potentials (top to bottom, −59 mV, −61 mV, −55 mV, −57 mV) illustrate depolarization (dpol) and hyperpolarization (hpol). (**B**) Responses of four different Purkinje cells to a visual stimulus (blue step, all panels) that did not evoke swimming. Dotted lines, top to bottom, −57 mV, −54 mV, −55 mV, −50 mV. (**C**) Responses of two different Purkinje cells to a visual stimulus that elicited swimming, either with (top) or without (bottom) complex spikes during the swimming episode. Dotted lines, top and bottom, −63 mV, −66 mV. (**D**) Responses of two different Purkinje cells to tactile stimulus (purple step, all panels), either with (top) or without (bottom) complex spikes during the swimming episode. Dotted lines, top and bottom, −60 mV, −66 mV. (**E**) Swim latency vs. climbing fiber response (complex spike) latency relative to stimulus onset for visual (blue), and tactile (purple) evoked swimming for all cells. Data for spontaneous (grey) swimming is included at a latency of 0. Dotted line, unity. (**F**) Mean latency of first climbing fiber response (complex spike) relative to swimming onset calculated from difference between x and y values in (**E**). Zero indicates coincidence. $F_{(2,80)}=1.84$. (**G**) As in (**E**) but for latency of long-lasting (>200 ms) pfEPSP-initiated depolarizations. (**H**) As in (**F**) but for long-lasting depolarizations. $F_{(2,88)}=16.54$. In all figures, data are plotted as mean ± SEM, and asterisks on plots indicate p<0.05.

Next, we measured the latency of Purkinje cell responses relative to the onset of swimming. On average, climbing fiber responses lagged the onset of spontaneous and sensory-evoked swimming (*Figure 2E and F*; spontaneous: 102 ± 31 ms, N = 109 episodes/29 cells; visual: 40.1 ± 23.4 ms, N = 23 episodes/23 cells; tactile: 70.3 ± 13.6 ms, N = 42 episodes/42 cells; p=0.17). The timing of complex spikes, however, had relatively high variance (spontaneous, coefficient of variation = 1.1; visual, CV = 1.1; tactile, CV = 1.1), such that complex spikes in some cells preceded swimming. This observation helps exclude the possibility that complex spikes simply report a visuomotor mismatch; if so, they would always lag swimming onset, and would be equally probable during all forms of swimming, neither of which was the case.

Because pfEPSPs were numerous and probably undersampled (see *Materials and methods*), we estimated the latency of substantial parallel-fiber drive by measuring the onset of long-lasting (>200

ms) depolarizations initiated by pfEPSPs, which drove simple spikes (*Figure 1I*). In contrast to climbing fiber responses, the timing of these events nearly coincided with spontaneous and visually evoked swimming onset (*Figure 2G*; lag for spontaneous: −6.3 ± 8.3 ms, N = 91 episodes/26 cells; visual: −23.4 ± 13.0 ms, N = 28 episodes/28 cells). With tactile stimulation, however, pfEPSPs significantly lagged swimming onset, by 52.9 ± 14.2 ms (*Figure 2H*, N = 35 episodes/35 cells, p<0.001). The high variance of response latencies suggests that Purkinje cells may play heterogeneous roles in spontaneous and sensory-evoked swimming, possibly ranging from triggering to reporting these motor responses (but see ablation studies below). These observations are consistent with those reported for Purkinje cells during the optomotor response (*Scalise et al., 2016*). More generally, the data demonstrate that individual Purkinje cells respond to a variety of sensory modalities, as well as to motor commands.

## A cerebellar learning task in the zebrafish

Next, to prepare to study Purkinje cell activity during associative learning, we tested whether larval zebrafish could be conditioned to produce fictive swimming in response to a visual cue. In these experiments, the ventral root signal was recorded without concurrent recording from Purkinje cells. A 2 s blue light that was low enough contrast not to evoke fictive swimming served as a conditional stimulus (CS; see *Materials and methods*). The CS was immediately followed by the tactile unconditional stimulus (US), which elicited the unconditional response (UR) of fictive swimming (*Figure 3A*). Because the start-to-start interval between individual trials was 40–55 s, the number of trials roughly corresponded to the number of minutes of training. With repeated presentations of the paired CS and US, a subset of fish developed a conditional response (CR) of fictive swimming to the low-contrast light (*Figure 3A and B*).

In mammalian studies of cerebellar associative learning, particularly eyelid conditioning, a 2 s CS is relatively long (e.g., *Garcia and Mauk, 1998*). We therefore tested briefer CS-US intervals. Fish trained with a 2 s CS produced CRs on 32.0 ± 5.0% of trials (N = 22). With a 1 s CS, performance was similar (29.1 ± 6.6%, N = 15). With a 0.5 s CS, fish produced CRs on about half as many trials as with a 2 s CS (17.8 ± 4.4%, N = 15), but performance was statistically indistinguishable with all intervals (p=0.39, *Figure 3B*). All subsequent experiments used a 2 s CS, which gave the largest number of fish producing a high fraction of CRs and which had the additional benefit of providing the longest window to detect changes in Purkinje cell and ventral root activity during the CS.

Because different fish varied considerably in the percentage of CRs they performed (*Figure 3B*), for analyses of response properties and of manipulations of learning, fish were separated into 'learners,' which performed CRs on >20% of trials in a 70 trials experiment (N = 10/22 fish; 45%), and 'non-learners' (N = 12/22; 55%), which did not. While non-learners performed CRs on only 4 ± 1% of trials, learners generated CRs on 44 ± 2% of trials (p<0.001). In addition, learners performed consecutive CRs (i.e., they 'acquired' CRs) after 27.5 ± 5 trials and reached a plateau of 59 ± 5% CRs after 40 trials (*Figure 3C*).

The CRs that emerged over training likely reflected associative learning rather than sensitization of visually evoked swimming, since unpaired presentations of the CS and US (pseudoconditioning) did not lead to the emergence of CRs (*Figure 3C*; N = 10 fish, p<0.001). To test whether CRs extinguished, seven additional fish received up to 30 paired presentations of the CS and US. Six fish achieved a criterion of 3–5 consecutive CRs in 16 ± 3 trials. Subsequent presentations of the CS alone extinguished CRs in all six fish after 5 ± 1 trials, providing further evidence that the swimming episodes during the visual stimulus after training were indeed associatively learned CRs.

To test whether the behavior reflected cerebellar learning, we ablated the cerebellum before training (N = 10). Without an intact cerebellum, fish still displayed fictive swimming both in response to high-contrast light and to the tactile stimulus (*Figure 3D*), indicating that the loss of the cerebellum did not disrupt sensory or motor pathways required for URs, nor did it abolish the ability to generate motor output to drive swimming. The fish did not acquire CRs, however (*Figure 3C*; p<0.001), confirming that the cerebellum is required for this form of learning.

In intact fish, the properties of CRs changed over repeated trials, with respect to the latency to the initiation of swimming, the number of motor 'bursts' within swimming episodes, and the frequency of these bursts. The initial CR occurred in the middle of the CS and was relatively brief and slow (latency, 1.1 ± 0.19 s; bursts/episode, 7.2 ± 1.6; burst frequency, 23 ± 1.4 Hz). Across subsequent trials, the changes in swimming properties reached a plateau by approximately the fifth CR

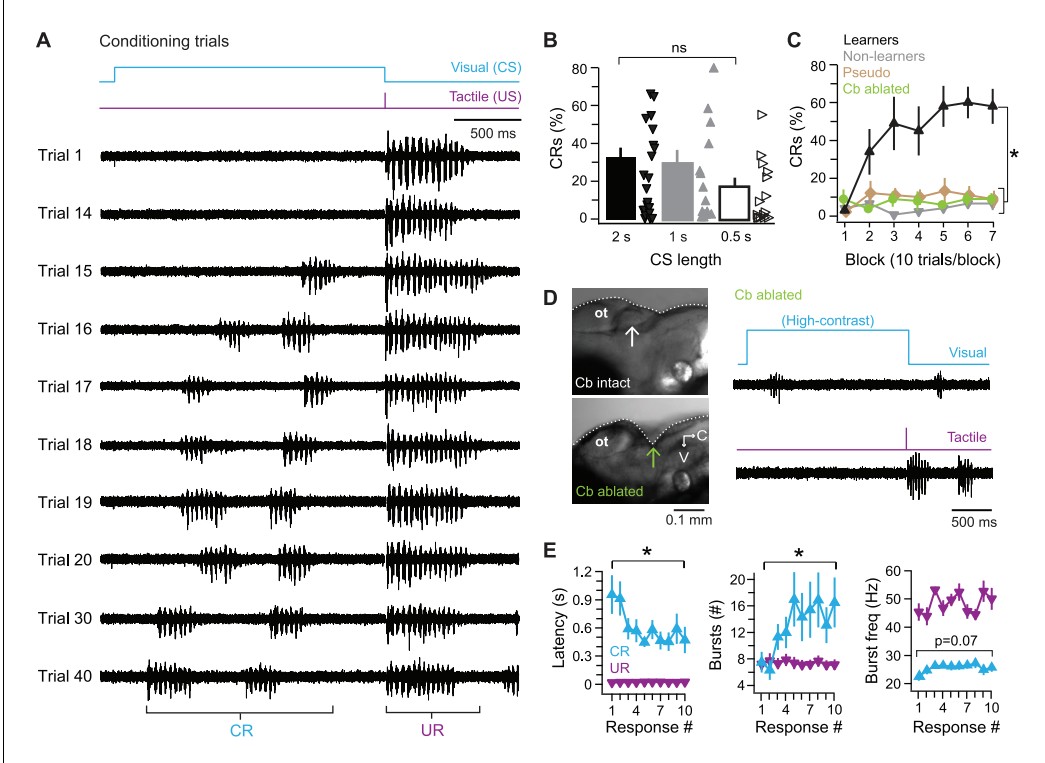

**Figure 3.** Cerebellar associative learning. (**A**) Sample ventral root recordings during training, illustrating the emergence and persistence of conditional responses (CRs) over time, as well as unconditional responses (URs). Trial numbers as indicated. Blue step, conditional stimulus (CS); purple step, unconditional stimulus (US). (**B**) Percentage of trials with a CR over all 70 trials. Bars are means; markers are data from individual fish; ns, not significant. $F_{(2,49)}$=0.97. (**C**) Percentage of trials with a CR per 10-trial block for learner and non-learner groups from fish in (**B**) trained with 2 s CS. $F_{(6,18)}$=4.92. (**D**) Left side view of exposed zebrafish brain before (top) and after (bottom) ablation of the cerebellum (arrow). V, ventral; C, caudal. Right, sample ventral root recordings of swimming evoked by a high-contrast visual stimulus (top) or tactile stimulus (bottom) after cerebellar ablation. (**E**) Changes in CR properties from first 10 CR trials in learner fish. UR data plotted for comparison. Left, swimming latency (relative to CS or US onset), CR: $F_{(9,81)}$=3.80, UR: $F_{(3.98,81)}$ = 0.78, p=0.5. Middle, number of bursts per swim response, CR: $F_{(9,81)}$=2.8, UR: $F_{(9,81)}$=1.60, p=0.13. Right, frequency of bursts, CR: $F_{(9,81)}$=2.84, UR: $F_{(3.47,81)}$ = 1.82, p=0.16.

trial; the latency decreased (p<0.001); the number of bursts increased (p=0.006); and swimming frequency increased slightly (p=0.07; *Figure 3E*; UR data overlaid for comparison). The observation that multiple parameters of learned swimming change during acquisition of the CR suggests that cerebellar circuits influence premotor regions that not only control the initiation of swimming but also its patterning (e.g., duration and frequency). In addition, the gradual approach of CR parameters to plateau values indicates that this form of learning is not all-or-none, but stabilizes over time.

## Purkinje cell activity during learning

To test whether and how Purkinje cell firing was modified during learning, we recorded from Purkinje cells during training in 31 fish that learned. In these fish, CRs were acquired in 12.4 ± 1.6 trials, and experiments lasted 20.7 ± 1.8 trials. The activity of all Purkinje cells changed over training, but the responses were heterogeneous across the population. Inspection of climbing fiber and parallel fiber responses that developed over the course of training suggested that Purkinje cells might be distinguished according to their patterns of complex spiking *after* learning had taken place. Specifically, they could be categorized as firing 0, 1, or >1 complex spike in association with each CR. It is worth emphasizing, however, that alternate or additional classification schemes are not ruled out by this approach. We considered the possibility of classifying Purkinje cells on the basis of parallel fiber

drive (pfEPSP-driven spiking), but the data did not fall into self-evident categories, and most criteria seemed arbitrary. We therefore proceeded with the preliminary classification of Purkinje cell responses based on complex spikes fired during the CR, which placed every cell unequivocally into one of three groups ('classes'), and tested its validity by further analysis. *Figure 4A–4C* illustrates sample traces of Purkinje cell responses, followed by schematics illustrating the responses of every cell in each group, from trials after fish produced at least two consecutive CRs.

The first group, multiple complex spike cells (MCS, N = 13/31), produced two or more complex spikes during the CR (*Figure 4A*). In these cells, complex spikes were evident on every trial that included a CR. pfEPSPs with simple spikes and/or hyperpolarization were present, but variable. The second group, single complex spike cells (SCS, N = 11/31), generated one complex spike during the CR on most trials (*Figure 4B*). This complex spike tended to be temporally associated with the swim episode, and could also be accompanied by pfEPSPs with simple spikes or by hyperpolarization. The third group, zero complex spike cells (ZCS, N = 7/31), produced no complex spikes during the CR on all CR trials, instead displaying summating parallel fiber pfEPSPs and simple spikes (*Figure 4C*). All ZCS cells did, however, fire complex spikes to the US (on 35 ± 10% of trials), so they were indeed Purkinje cells innervated by climbing fibers with task-related activity. By comparison, all MCS cells also produced complex spikes to the US (on 67 ± 7% of trials), while 9 of 11 SCS cells produced complex spikes to the US (on 46 ± 7% of trials). Simple spike rates at the beginning of recording did not differ between cell types (MCS: 3.4 ± 1.2 Hz; SCS: 9.3 ± 2.4 Hz; ZCS: 5.6 ± 2.7 Hz; One-way ANOVA: $F_{(2,18)}$=2.12, p=0.15).

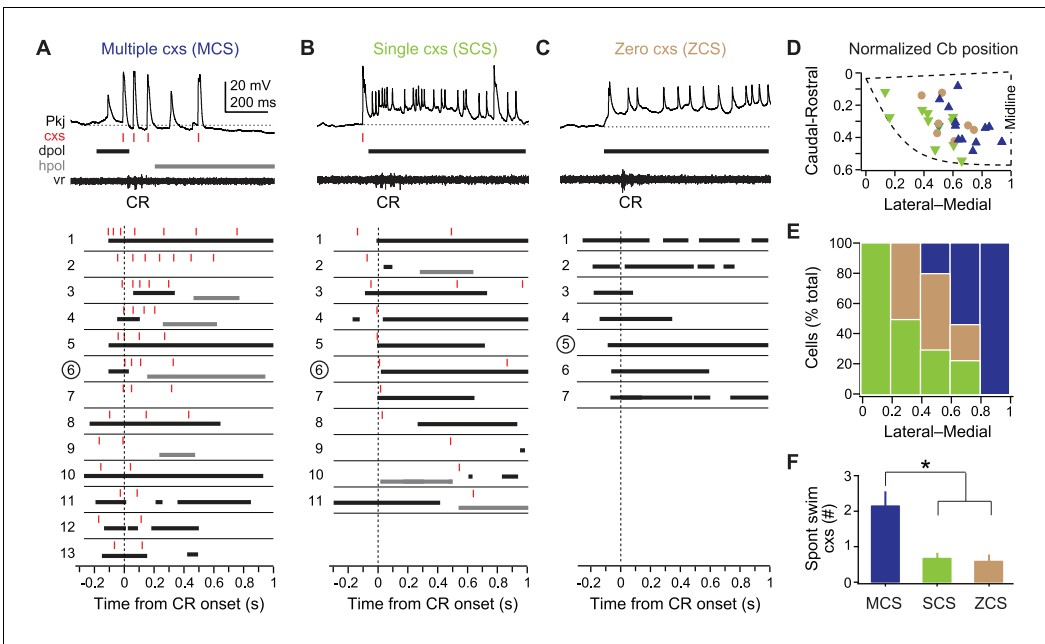

**Figure 4.** Three classes of Purkinje cell activity during learned swimming. (A) Sample recording from a multiple complex spike (MCS) cell, top, during the conditional response (CR) late in training. Horizontal dotted line, −55 mV. Schematized responses from MCS cells, below, aligned to the CR onset (vertical dotted line). For (A), (B), and (C): red ticks, complex spikes; black bars, pfEPSP-initiated depolarizations (dpol); grey bars, hyperpolarizations (hpol). MCS cells are ordered by the number of complex spikes within the CR. The number corresponding to the sample recording is circled. (B) As in (A) but for single complex spike (SCS) cells. Horizontal dotted line, −59 mV. SCS cell schematized responses are ordered by the latency of CR-related complex spikes. (C) As in (A) but for zero complex spike (ZCS) cells. Horizontal dotted line, −56 mV. ZCS cell schematized responses are ordered by the latency of CR-related pfEPSPs. (D) Topographical distribution of MCS, SCS, and ZCS cells in the cerebellum. The position of the rostrolateral, rostromedial, and caudomedial corners are plotted (dashed line) to approximate the edges of the hemisphere, and relative positions of cells were calculated accordingly. (E) Ratios of each class of Purkinje cells along the mediolateral cerebellar axis. (F) Number of complex spikes in each class of Purkinje cells during episodes of spontaneous swimming. $F_{(2,22)}$=7.78.

We then tested whether this categorization provided a reasonable classification of distinct groups of Purkinje cells for this associative learning task. Plotting the location of cells coded by group revealed that these neurons were topographically ordered along the mediolateral axis. MCS cells predominated most medially and were absent from the most lateral zone, SCS cells predominated most laterally and were absent from the most medial zone, and ZCS cells lay only between these extremes (*Figure 4D and E*). Next, we examined the activity of these cells during spontaneous swimming, before learning had occurred. This analysis showed that the probability and number of complex spikes that occurred during spontaneous swimming was partially predictive of the classification of Purkinje cells after learning; specifically, of the 10 cells that fired at least one complex spike on every episode, eight became MCS cells, resulting in a larger mean number of complex spikes during spontaneous swimming for this group (*Figure 4F*; p<0.02). Because all these analyses taken together provided reasonable anatomical and physiological support for the initial classification scheme, we next analyzed each group separately for changes in Purkinje cell activity over the course of training.

## Multiple complex spike cells

For MCS cells, we examined complex spikes during (1) spontaneous swimming, (2) UR swimming, and (3) the CS over repeated trials until CR swimming emerged (*Figure 5A and B*). MCS cells produced more complex spikes during CRs than during either spontaneous swimming or URs (p<0.02; *Figure 5C*, *top*). Also, the initial complex spike associated with each event approximately coincided with spontaneous swimming onset and consistently lagged the UR, but *preceded* the CR (p<0.02; *Figure 5C*, *bottom*, *Figure 5D*). In addition to firing complex spikes, most MCS cells (N = 9/13) showed pfEPSPs and simple spikes during CRs, while the remainder produced hyperpolarizations (N = 4/13; *Figure 5B*).

In MCS cells, the quantity and timing of complex spikes throughout the visual stimulus (i.e., not only during the CR) was plastic over successive trials (*Figure 5B*). Over the course of training, the number of complex spikes increased in MCS cells (p<0.001). Most of these events occurred in the first 500 ms of the CS, and their number increased gradually over training in MCS cells (p<0.03). No such change occurred in SCS cells, ZCS cells, or cells from pseudoconditioned fish (*Figure 5E*).

To investigate whether the learning-associated complex spikes in MCS cells were related to the motor command for the CR or the sensory input of the CS, we took advantage of the trial-to-trial variability in performance and analyzed *only* those trials that lacked CRs. The trials were grouped as *early* (three trials at the beginning of training), *middle* (three trials just before expression of consecutive CRs), and *late* (three trials just after expression of consecutive CRs). Notably, even when learned swimming responses were absent, MCS cells fired more complex spikes during the first 500 ms of the CS on middle and late trials than during early trials (p<0.04; *Figure 5F*, *left*). In contrast, the number of complex spikes fired during URs tended to stay constant or decrease in MCS cells (p=0.09, middle vs. late; *Figure 5F*, *right*).

Examining the temporal relationship between the CS and complex spikes further indicated that complex spikes were generated in response to sensory input. Trial-by-trial analysis revealed that the latency of the first complex spike tended to decrease over training (p=0.07; *Figure 5G*) and became more precisely timed to the CS onset (*Figure 5H*; CV, *early:* 0.54 ± 0.08; CV, *late:* 0.19 ± 0.04, p<0.001). The complex spike in late trials was better timed to the CS than to the CR (CV, *CR:* 0.57 ± 0.06; p<0.002). These results suggest that, during learning, climbing fibers that are responsive to the sensory CS undergo changes, which preferentially affect MCS Purkinje cells.

## Single complex spike cells

All 11 SCS cells showed changes in activity associated with spontaneous swimming, URs, and CRs (*Figure 6A and B*). Eight SCS cells responded with increases in simple and complex spikes, one cell hyperpolarized, and two cells showed both excitatory and inhibitory responses. Complex spikes during spontaneous swimming and URs followed swimming onset with lags of 96.3 ± 42 ms and 98.5 ± 33 ms (*Figure 6C*). In contrast, complex spikes occurred either just before CR onset (latency = −74 ± 17 ms; N = 40/53 CRs) or after CRs ended (223 ± 28 ms, N = 13/53 CRs), suggesting an association with the learned motor response (*Figure 6B and C*). A number of observations appear consistent with this idea. First, in SCS cells, unlike MCS cells, the complex spike latency was more precisely timed to either CR onset (CV = 0.49 ± 0.08) or offset (CV = 0.35 ± 0.13) than to the CS

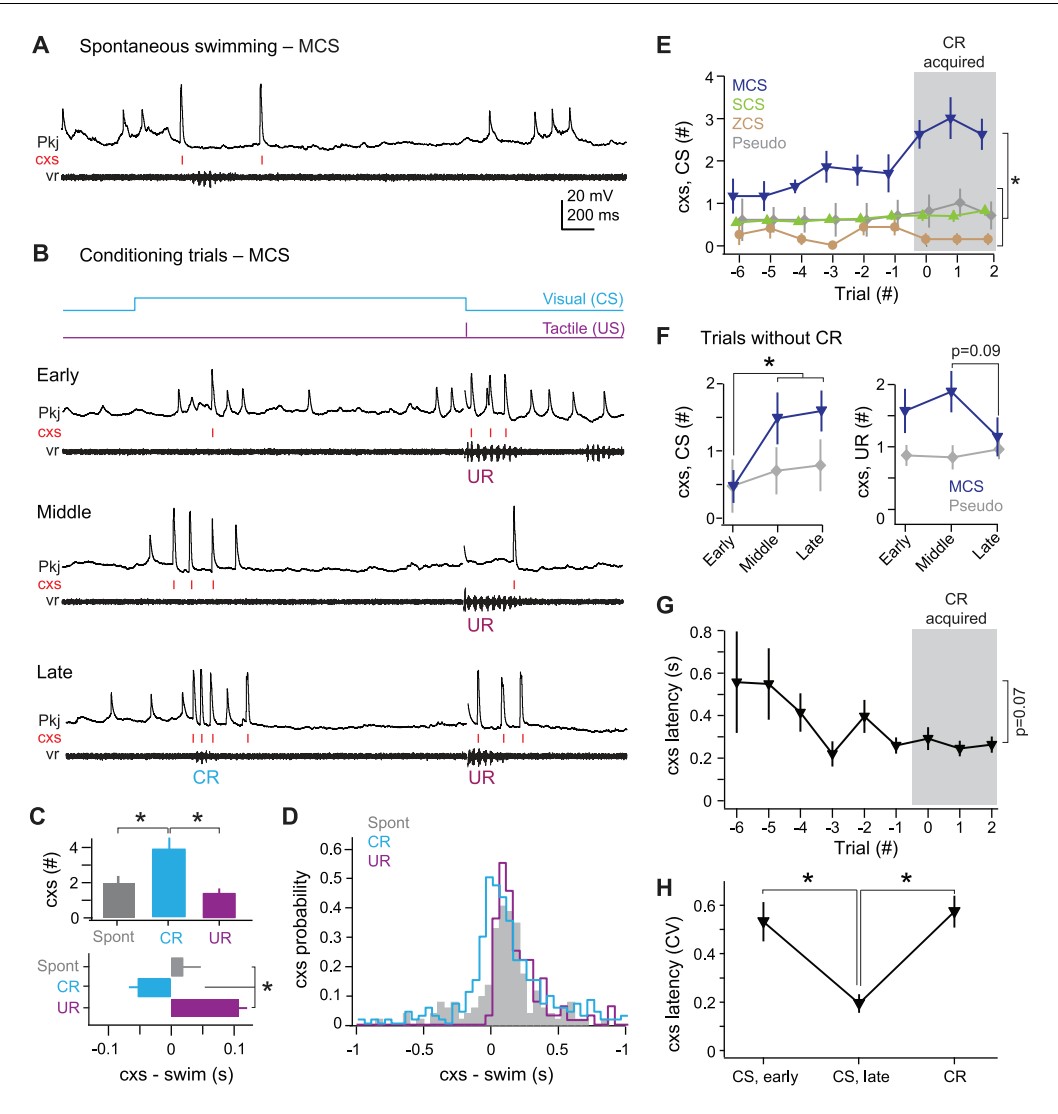

**Figure 5.** MCS Purkinje cell responses over the course of cerebellar learning. (**A**) Sample recording from an MCS cell during spontaneous swimming. (**B**) Sample recording from the same MCS cell in (**A**), at the beginning of training (early), just before acquisition of the conditional response (CR; middle), and after CR acquisition (late). (**C**) Top, number of complex spikes during three forms of swimming: spontaneous, the CR, and the unconditional response (UR). $F_{(2,35)}=9.20$. Bottom, mean latency of the first complex spike in all cells. $F_{(2,35)}=23.69$. (**D**) Distribution of the latency of all complex spikes during swimming in all MCS cells (bin width = 50 ms). (**E**) The number of complex spikes during the first 500 ms of the conditional stimulus (CS). MCS, SCS (single complex spike), and ZCS (zero complex spike) cells, and cells from pseudoconditioned fish are shown for comparison. $F_{(24,248)}=1.95$. For all similar plots, trial 0 is the first of consecutive trials with CRs for trained fish, or the tenth trial for pseudoconditioned fish (i.e., the median acquisition trial for trained fish). (**F**) The number of complex spikes during three trials *without* a CR for early, mid, and late training, for the first 500 ms of the CS (left; $F_{(1.76,32)} = 6.47$) and 500 ms after the US (right). Cells from pseudoconditioned fish are shown for comparison. (**G**) Mean complex spike latency relative to CS onset for MCS cells. $F_{(8,40)}=2.01$. (**H**) CV of complex spike timing relative to CS onset early in training, late in training, and relative to the CR onset for MCS cells. The CV was calculated across trials for each cell, and the mean CV for all cells is plotted. CS points include trials regardless of whether a CR was produced. CR point represents the latency of the first complex spike.

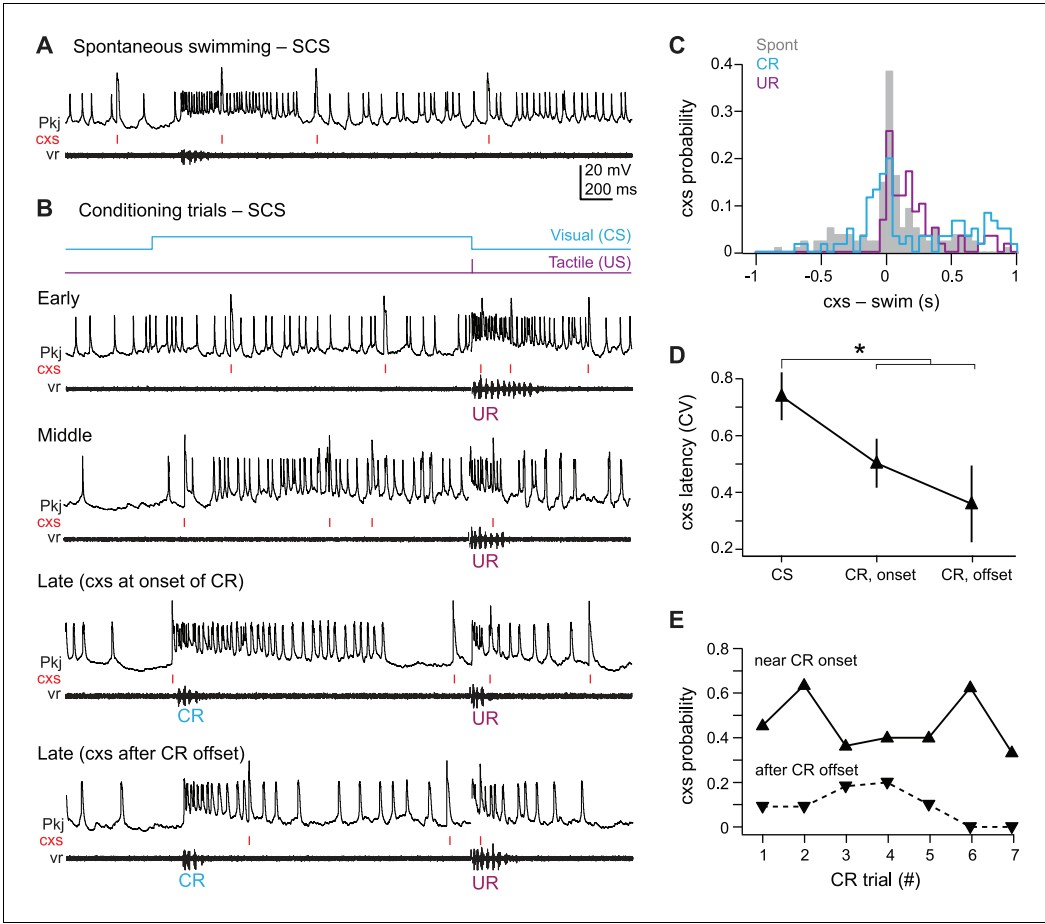

**Figure 6.** SCS Purkinje cell responses over the course of cerebellar learning. (**A**) Sample recording from a single complex spike (SCS) cell during spontaneous swimming. (**B**) Sample recording from the same SCS cell in (**A**) at the beginning of training (early), just before acquisition of the conditional response (CR; middle), and on two trials after CR acquisition with complex spike activity either near CR onset or after CR offset within 300 ms of the CR. (**C**) The distribution of the latency of all complex spikes in SCS cells relative to spontaneous, CR, and unconditional response (UR) swimming onset. (**D**) The coefficient of variation (CV) of complex spike (cxs) latency relative to the conditional stimulus (CS) onset, the CR onset, and the CR offset for all SCS cells. The CV for each cell was calculated across trials, and the mean CV for all cells is plotted. (**E**) The probability of a complex spike within 300 ms of the onset (solid lines) or within 300 ms after the offset (dashed lines) of the CR for all SCS cells.

(CV = 0.71 ± 0.08; p=0.012, paired t-test; *Figure 6D*). Also, the probability of a complex spike was relatively high near CR onset, while the probability of complex spikes occurring within 300 ms of the end of the CR transiently increased on the third and fourth CR trials (*Figure 6E*). Additionally, a disproportionate number of SCS cells did not respond to the visual stimulus before training (46% of SCS cells *vs.* 23% of all 49 cells tested, including those from pseudoconditioned and untrained fish), and the number of CS-related complex spikes in SCS cells did not change consistently over training (*Figure 5E*). Together, these data support the idea that climbing fiber input to SCS cells is more directly related to the motor command for the CR than the sensory input of the CS.

## Zero complex spike cells

All 7 ZCS cells showed many pfEPSPs, which frequently elicited long-lasting depolarizations with simple spikes. These events were associated with spontaneous swimming, the CS on non-CR trials, and CRs (*Figure 7A and B*). Over the course of training, the total number of pfEPSPs during the CS increased even before CRs were expressed and reached a maximum after CRs emerged (p<0.05). In contrast, the number of pfEPSPs did not increase in MCS cells, SCS cells, and cells from

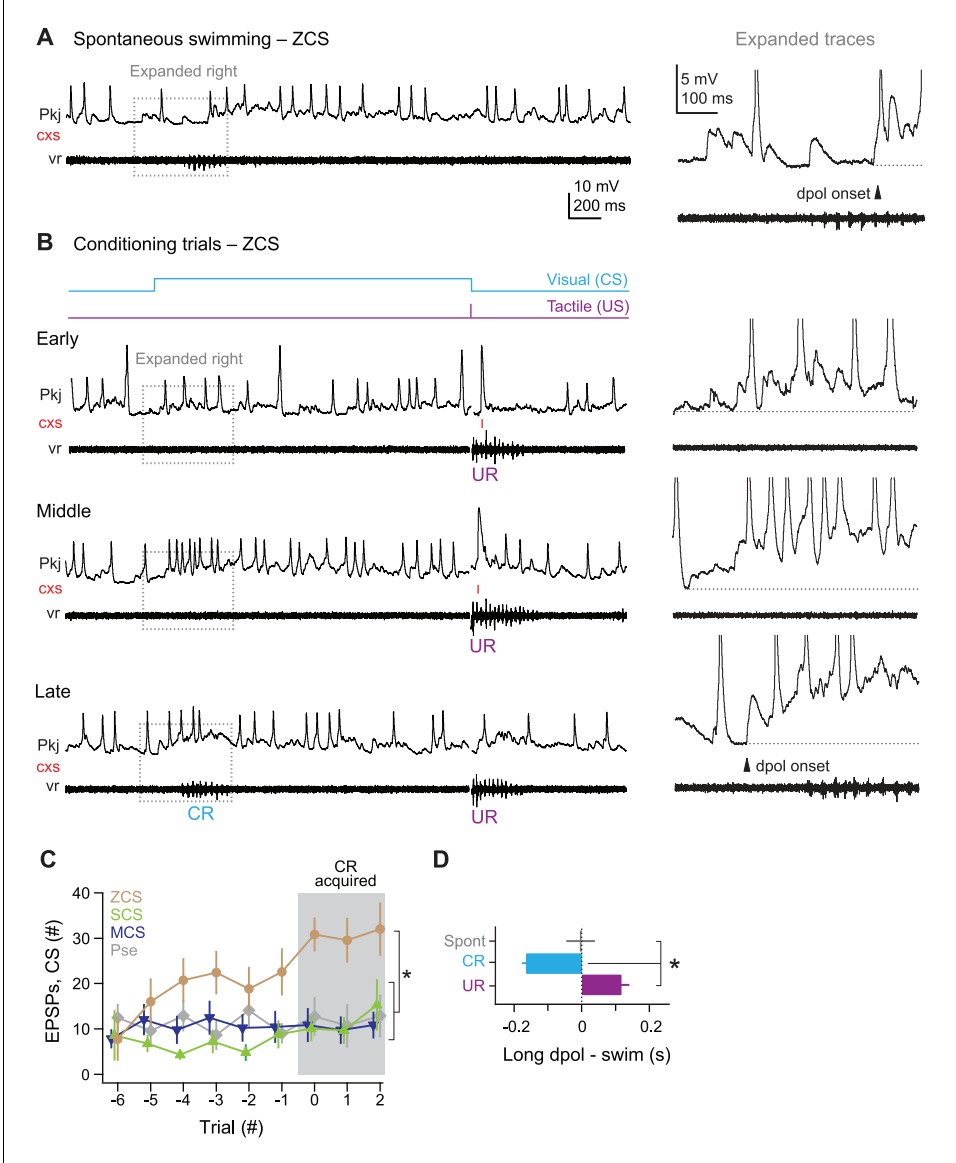

**Figure 7.** ZCS Purkinje cell responses over the course of cerebellar learning. (**A**) Left, sample recording from a zero complex spike (ZCS) cell during spontaneous swimming. Right, magnification of boxed region at left. Arrow: onset of long-lasting depolarization. Dotted line, −51 mV. (**B**) Sample recording from the same ZCS cell in (**A**) at the beginning of training (early), just before acquisition of the conditional response (CR; middle), and after CR acquisition (late). Dotted lines, top to bottom, −51 mV, −54 mV, and −54 mV. (**C**) The number of parallel-fiber EPSPs (pfEPSPs) during the 2 s conditional stimulus (CS) for ZCS cells, as well as MCS, SCS cells and cells from pseudoconditioned fish. $F_{(11.16,198)}$ = 2.18. (**D**) Latency of pfEPSP-initiated long-lasting depolarization relative to the onset of spontaneous swimming, the CR, and the unconditional response (UR). $F_{(2,18)}$=22.33.

pseudoconditioned fish (*Figure 7C*). The onset of long-lasting (>200 ms) depolarizations nearly coincided with spontaneous swimming (latency, 2 ± 36 ms) and lagged tactile-evoked swimming (59 ± 19 ms). Long-lasting depolarizations, however, significantly preceded learned swimming (lag, −171 ± 15 ms, p<0.001 *Figure 7B and D*). Together, these observations suggest that the CS-associated parallel fiber drive to ZCS cells increases over training.

## Suppressing Purkinje cell simple spikes

The three categories of Purkinje cells showed distinct responses, but together they provided evidence that both parallel fiber EPSP-driven simple spikes and climbing fiber-mediated complex spikes are systematically modified during associative learning. In addition, since some changes precede learning whereas other changes continue to develop after CRs emerged, different components of Purkinje cell activity likely contribute differentially to acquisition (learning the association of paired stimuli during training), expression (generation of learned motor responses), and maintenance (retaining the learned association and continuing to produce learned movements). Although at present we have no reliable method to control the activity of only one group of Purkinje cells at a time, we reasoned that we could begin to dissect the roles of simple and complex spikes, and possibly infer roles of the cell classes, by optogenetically interfering with activity of all Purkinje cells.

To do so, we used a transgenic fish line in which Archaerhodopsin-3 ('Arch') was expressed only in Purkinje cells (*Matsui et al., 2014*; see also *Figure 1A*). Arch-activating light, which was of higher intensity and different wavelength than the light used as a CS (see *Materials and methods*), was directed onto the cerebellum through the microscope objective and constrained to the minimal diameter necessary to illuminate the cerebellum fully. Voltage-clamp recordings from Purkinje cells showed that Arch activation evoked an outward current of $18.6 \pm 2.7$ pA (N = 6; *Figure 8A*, *top*) that reached a maximum within 5 ms. This current hyperpolarized current-clamped Purkinje cells by $27.1 \pm 3.4$ mV and greatly suppressed simple spikes, from $6.0 \pm 1.8$ spikes/s to $0.4 \pm 0.3$ spikes/s (p=0.008; *Figure 8A bottom, Figure 8B*). Consistent with the large amplitude of climbing fiber EPSCs, however, complex spikes persisted during Arch activation ($0.16 \pm 0.02$ spikes/s, *Figure 8B*). These rates were comparable to control ($0.12 \pm 0.04$ spikes/s; p=0.6), suggesting that complex spikes were not indirectly suppressed via olivocerebellar loops (e.g., *Medina et al., 2002*).

To evaluate the effect of the Arch-dependent outward current during substantial excitatory drive, we recorded Purkinje cell activity during episodes of learned swimming (N = 2 MCS cells and 1 SCS cell). With Arch activation, simple spikes during swimming were largely suppressed, but summating pfEPSPs and complex spikes remained (*Figure 8C*). Importantly, this experiment also demonstrated that learned swimming could occur during Arch activation, indicating that the behavior did not rely solely on Purkinje cell simple spiking (analyzed further below). We therefore concluded that Arch could reasonably be used to suppress Purkinje cell simple spikes preferentially, without affecting complex spikes.

We therefore applied Arch-activating illumination at various points during training. In all experiments, Arch-activating light was applied *only* during the presentation of the CS, so that responses to the US and any other signals not overlapping with the CS could proceed unperturbed. Since these experiments required many trials, Purkinje cell recordings were omitted to maximize the number of fish from which complete data sets could be obtained. Nevertheless, the previous experiments made it possible to infer the effects on different cell groups: ZCS cells, in which synaptically driven simple spikes occur throughout acquisition and expression, are expected to be most affected by such a manipulation, while the subset of MCS and SCS cells that increase their simple spiking during swimming would become affected after CRs emerge.

## Simple spike suppression during acquisition

We first tested whether suppressing simple spikes affected the acquisition of CRs. Naïve Arch[+] (N = 49) and control (N = 47) fish were trained with paired presentations of the CS and US and pseudoconditioned fish received unpaired stimuli (N = 43, of which 14 were Arch[+]). Among control fish, conditioning with the additional Arch-activating illumination proceeded as it did with the CS light alone (*Figure 8D*, *left*). The CR percentage increased significantly, from $9.2 \pm 2.9\%$ to $28.7 \pm 5.1\%$ (p<0.001), and 20/47 fish (42.6%) reached the criterion of producing CRs on >20% of trials (*Figure 8D*, *right*). In contrast, in Arch[+] fish, the CR percentage went from $6.1 \pm 2.2\%$ only to $12.6 \pm 3.7\%$ (*Figure 8D*, *left*; p=0.17); this change was smaller than in control (p=0.003) and indistinguishable from pseudoconditioned fish (p=0.7). When compared to control fish, fewer than half as many Arch[+] fish could be classified as learners (N = 9/49, 18.4%, p=0.014, *Figure 8D*, *right*).

This reduced CR probability may have resulted either because CRs were not acquired, owing to a disruption of plasticity, or because CRs could not be expressed, even with normal development of plasticity. To distinguish between these possibilities, we tested the effect of restoring simple spikes

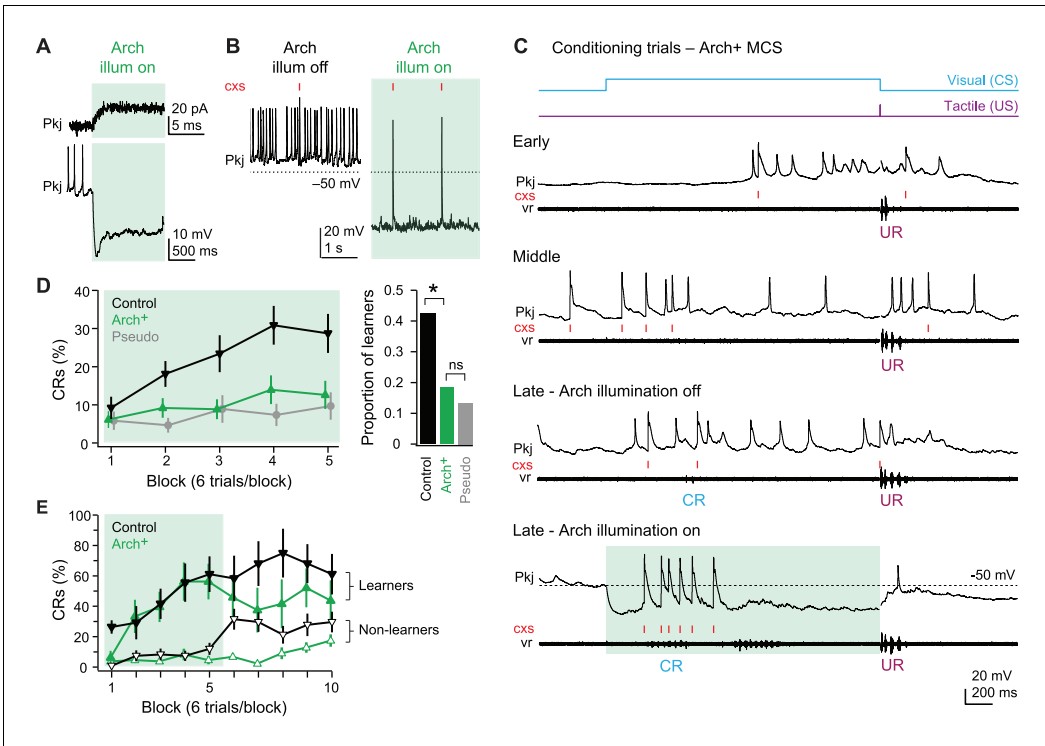

**Figure 8.** Effects of Arch-mediated simple spike suppression on acquisition of learned responses. (**A**) Sample voltage-clamp (top), holding potential = −60 mV, and current-clamp (bottom) recordings from an Arch[+] Purkinje cell during Arch activation. Green shading in all panels in all figures indicates cerebellar illumination with Arch-activating light. (**B**) Current-clamp recording of an Arch[+] Purkinje cell without (left) and with (right) activation of Arch. Two complex spikes (cxs) are evident during illumination. (**C**) Response of an Arch[+] multiple complex spike (MCS) Purkinje cell at the beginning of training (early), just before acquisition of the conditional response (CR; middle), after CR acquisition without simple spike suppression (late - Arch illumination off) and after CR acquisition with simple spike suppression (late - Arch illumination on). (**D**) Left, CR percentage per 6-trial block of control, Arch[+], and pseudoconditioned fish that received Arch-activating cerebellar illumination during visual stimulation from the onset of training. $F_{(8,544)}=2.14$. Right, proportion of learner fish for each group. $X^2_{(1, N=96)}=6.66$. (**E**) CR percentage per 6-trial block for Arch[+] (green) or control (black) fish classed as learners (closed symbols) or non-learners (open symbols) in the first 30 trials.

after training during Arch activation. For control and Arch[+] fish (N = 30 per group), 30 trials of training with Arch activated were followed by 30 trials in which the Arch-activating light was presented but displaced from the cerebellum. We reasoned that if CRs were not acquired, then CR probability would remain depressed after restoration of simple spikes. If, however, CRs simply could not be expressed, then CR probability would increase immediately upon simple spike restoration. For this analysis, fish were classified as learners and non-learners.

We first compared the *non-learner* fish in the control (18/30) and Arch[+] (22/30) groups. By definition, both groups produced a low proportion of CRs at end of the first 30 trials (control, 12.0 ± 4.0%; Arch[+], 4.5 ± 2.7%; *Figure 8E, open symbols*). When the Arch-activating light was displaced to the front of the fish, however, control fish immediately produced CRs on 32 ± 7% of trials, possibly owing to an increase in the intensity of the visual cue; this result suggests that, in some control fish classed as non-learners, the association of the CS with the US may have indeed been learned but the low-contrast CS fell below detection threshold. In contrast to controls, however, non-learner Arch[+] fish performed CRs on only 6.8 ± 2.1% of trials immediately after displacement (p<0.001 vs. control), revealing a real failure to learn in the first 30 trials.

We next compared the control (N = 12/30) and Arch[+] (N = 8/30) *learner* fish. Although a smaller proportion of Arch[+] fish learned CRs, acquisition proceeded similarly between the two groups (*Figure 8E, filled symbols*). At the end of 30 trials with Arch activation, CR probability was

indistinguishable (53 ± 12% for Arch$^+$, 58 ± 11% for control block 5; p=0.8). After displacement of the Arch-activating light, control fish continued to perform CRs at or above the level attained just before displacement (66 ± 6% for the last 30 trials). In contrast, with restoration of simple spikes, the CR probability in trained Arch$^+$ fish tended to decrease (*Figure 8E*, *filled symbols*) and did not improve through the rest of the session (42 ± 13% for the last 30 trials; p=0.10 vs. control). Thus, although some Arch$^+$ fish learned, the underlying cerebellar plasticity apparently adapted to the reduced level of simple spiking, such that restoring this activity generated a mild deficit in performance. Together, these results provide evidence that Purkinje cell simple spikes participate in the acquisition of learned responses. Specifically, reducing simple spiking alters cerebellar plasticity during learning and affects performance of this task.

## Simple spike suppression following CR acquisition

Next, we investigated whether simple spikes play a measurable role *after* learning has occurred normally. Because multiple parameters of learned swimming change after initial expression (*Figure 3E*), we reasoned that fish just beginning to produce CRs and fish producing CRs 'reliably' (i.e., on several consecutive trials) may differ in their sensitivity to simple spike suppression. To test this idea, control and Arch$^+$ fish were trained with simple spikes unperturbed until fish reached a pre-set learning criterion of generating CRs on 1, 3, or 6 consecutive trials. Next, the Arch-activating light was applied to the cerebellum for 10 trials and the persistence of CRs was measured. Regardless of the learning criterion, the CR latency, duration, swimming frequency, and amplitude recorded from the ventral roots did not differ between control fish and Arch$^+$ fish with simple spikes suppressed (unpaired t-tests, all p-values>0.25), and Arch$^+$ fish with and without simple spikes suppressed (paired t-tests, all p-values>0.30).

In control fish, the CR probability after the first CR was stable across these 10 trials, averaging 58 ± 13% (N = 20). In contrast, in the 1-CR Arch$^+$ group, which underwent simple spike suppression after a single CR (N = 20), the probability of a CR fell to 41 ± 13% over the first 4 trials of Arch activation (p=0.078 vs. control of 63 ± 8%) before recovering to control levels (p=0.8; *Figure 9A*, *top*). The Arch$^+$ groups that underwent suppression after 3 or 6 consecutive CR trials, however, showed no deficits in the likelihood of CRs (N = 36 in each group; *Figure 9A*, *middle and bottom*). In all cases, the mean CR probability across trials stayed near 60% (3-CR control, 60 ± 6%; 3-CR Arch$^+$, 61 ± 6%; 6-CR control, 65 ± 6%; 6-CR Arch$^+$, 70 ± 5%), consistent with the plateau of performance measured in initial behavioral experiments. Comparing the average response probabilities of the first five trials across all levels of training confirmed that simple spike suppression significantly impaired performance in the 1-CR Arch$^+$ group (p=0.02). Thus, only for nascent CRs do Purkinje cell simple spikes contribute to expression.

A remaining question, however, is whether simple spikes continue to play any detectable role after CRs are acquired, i.e., in maintenance of the learned response. To explore this possibility, we tested whether a period of simple spike suppression in well-trained fish influenced performance after simple spike restoration. Fish were trained to a 3-CR or 6-CR level with simple spikes intact (N = 12 in each group). Simple spikes were then suppressed during the CS for 5, 10, or 20 trials, after which they were restored for 10 more trials. In the 3-CR and 6-CR groups, learned swimming continued throughout the trials with the Arch-activating light (*Figure 9B and C*, *shaded trials*), with no difference in CR percentage between control and Arch$^+$ fish (all conditions p>0.3) or between the first and second half of the Arch-activation trials (all conditions p>0.3). After the Arch-activating light was displaced, the mean CR percentage also remained above 60% in control fish (*Figure 9B and C*, *boxed trials, Figure 9D*). In contrast, in Arch$^+$ fish subject to suppression trials, restoring simple spikes reduced the CR percentage in the 3-CR criterion group (*Figure 9B*, *boxed trials*). This deficit was greater with longer periods of reduced simple spiking. After 5, 10, or 20 suppression trials, the CR percentage fell from 58 ± 12% (p=0.3 vs. control), to 43 ± 9% (p=0.02) to 32 ± 10% (p=0.03; *Figure 9D top*).

Fish trained to the 6-CR criterion, however, showed no decrease in CR probability after simple spiking was restored, regardless of the number of trials with Arch activation (*Figure 9C*, *boxed trials, Figure 9D bottom*; p>0.5, all comparisons). These results further demonstrate that different degrees of learning, based on the reliability of CRs, have different sensitivities to simple spiking. Even fish performing with moderate reliability continue to adapt to a reduction in the level of simple

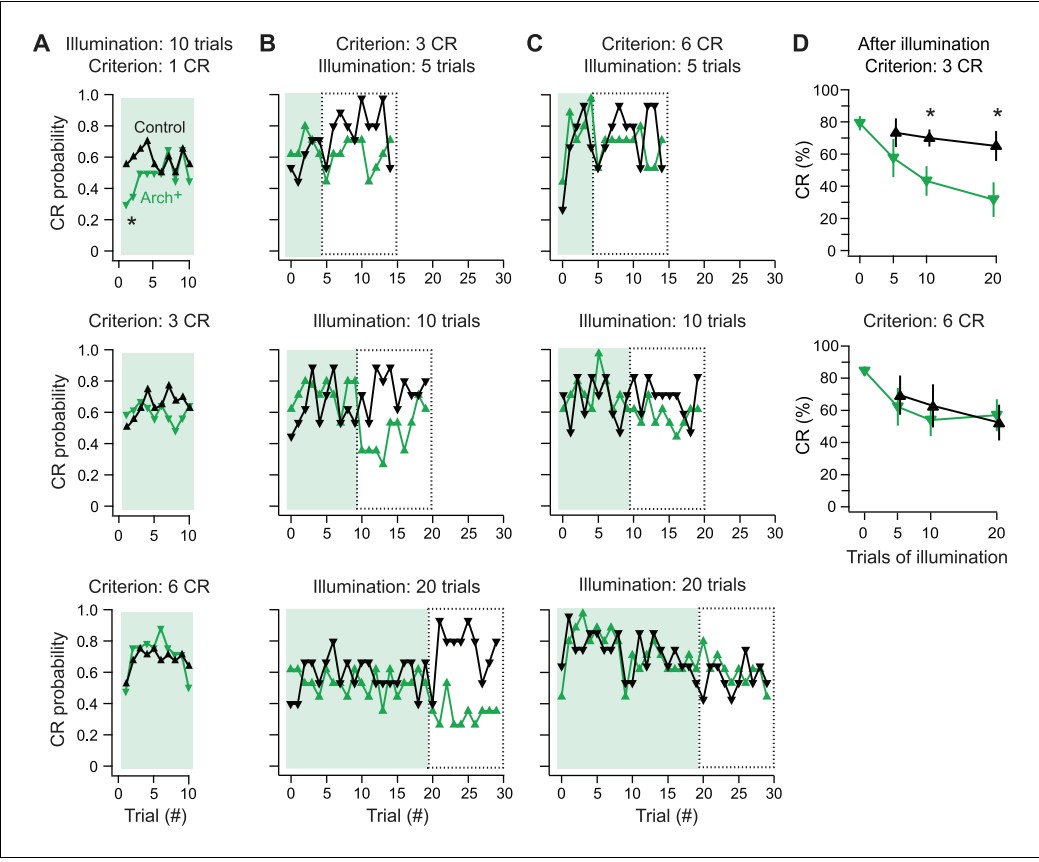

**Figure 9.** Effects of Arch-mediated simple spike suppression on expression and maintenance of learned responses. (A) Conditional response (CR) probability across fish during cerebellar Arch-activating illumination applied after fish reached a learning criterion of 1 (top), 3 (middle), or 6 (bottom) consecutive trials with a CR without simple spike suppression. (B) CR probability across fish during 5 (top), 10 (middle), or 20 (bottom) trials of Arch-activating illumination, followed by 10 trials without, for 3-CR fish. Dotted boxes, trials from which CR percentage is calculated in (D). (C) Same as (B) for 6-CR fish. (D) CR percentage calculated from trials after Arch-activating illumination for 3-CR (top) and 6-CR (bottom) fish (from dotted boxes in B and C).

spikes, such that when simple spiking is elevated, performance is degraded. Only in fish performing with high reliability does learned swimming persist independently of Purkinje simple spiking.

Together, these results suggest that the cerebellar plasticity required for CRs relies on Purkinje cell simple spikes during the CS, but this dependence does not segregate into straightforward categories. First, acquisition is facilitated by normal simple spiking; with simple spikes suppressed, most fish fail to learn. Those few that do acquire CRs remain particularly sensitive to restoration of simple spikes. Second, expression is facilitated by normal simple spiking only for nascent CRs; once CRs are reliably produced, normal simple spikes are no longer necessary for expression. Third, maintenance of CRs is largely independent of continued simple spiking during performance; however, only robustly learned responses (6-CR criterion) are maintained consistently throughout alterations in simple spike activity, whereas moderately learned responses (3-CR criterion) remain sensitive to further changes in simple spiking.

## Discussion

Here, we have made intracellular recordings from Purkinje cells during pre-training, acquisition, expression, and maintenance of cerebellum-dependent learned motor responses in larval zebrafish. The results indicate that distinct groups of Purkinje cells differ in both their location in the cerebellum and their synaptically driven electrophysiological responses through an associative learning task.

During training, complex spikes from climbing fiber activity become associated with sensory input from the CS in MCS cells, but with the motor command related to the CR in SCS cells. In ZCS cells, parallel fiber input increases during learning, but complex spikes do not change. Thus, different climbing fiber afferents convey distinct information to separate populations of Purkinje cells that all participate in this form of cerebellar learning. The results also demonstrate that Purkinje cell simple spikes play changing roles during different phases of this learning task. They strongly influence the acquisition of learned responses, transiently influence the expression of these responses, and become unnecessary for maintenance of well-learned behaviors.

## Pre-training responses of Purkinje cells

Consistent with the fact that 6–8 dpf larvae are free swimming and must encode and process sensory and motor information to survive, the Purkinje cell recordings made here from untrained fish illustrate that multimodal sensory signals are present relatively early in zebrafish development. Individual Purkinje cells fired simple and/or complex spikes to both light and tactile stimuli, as well as to episodes of fictive swimming, consistent with previous studies of larval zebrafish (*Hsieh et al., 2014*; *Sengupta and Thirumalai, 2015*; *Scalise et al., 2016*). The sources of multimodal cerebellar inputs are not yet identified in zebrafish; however, this range of responses is consistent with data from mammalian granule cells, which converge on Purkinje cells and respond to stimuli of different sensory modalities (*Azizi and Woodward, 1990*; *Chabrol et al., 2015*; *Ishikawa et al., 2015*). Likewise, mammalian inferior olivary neurons respond to sensory stimuli of different modalities and/or to movements (*Bauswein et al., 1983*; *Gellman et al., 1983*; *Kim et al., 1987*; *Winkelman et al., 2014*; *Ohmae and Medina, 2015*). Similar pathways likely exist in zebrafish, given the conservation of olivo-cerebellar circuitry across vertebrates (*Hodos and Butler, 1997*; *Takeuchi et al., 2017*).

## Topographical organization of the zebrafish cerebellum

Here, we categorized zebrafish Purkinje cells based on the number of complex spikes fired during learned swimming. This approach distinguishes the present study from past characterizations of Purkinje cells in naïve zebrafish (*Hsieh et al., 2014*; *Sengupta and Thirumalai, 2015*; *Scalise et al., 2016*). The classification reveals a relationship between learned activity and Purkinje cell position along the mediolateral axis. The topographic organization of Purkinje cells with different learned complex spike responses suggests that the larval zebrafish cerebellum is likely to be organized into olivocerebellar modules, as in other vertebrates (*Ruigrok, 2011*; *Cerminara and Apps, 2011*). In the mammalian cerebellum, separate regions of the inferior olive innervate zones of Purkinje cells that alternate in their expression of zebrin II (aldolase C). In addition, Purkinje cells with distinct intrinsic electrophysiological properties can often be distinguished by the presence or absence of zebrin II (*Brochu et al., 1990*; *Wadiche and Jahr, 2005*; *Kim et al., 2012*; *Zhou et al., 2014*; *Cerminara et al., 2015*). In zebrafish, however, zebrin II and other candidate biomarkers are expressed throughout the Purkinje cell population, suggesting that other molecular signals are involved in setting up olivocerebellar modules in zebrafish (*Meek et al., 1992*; *Bae et al., 2003*; *Takeuchi et al., 2017*).

The present data also add to previous findings of organized anatomical and functional heterogeneity in the larval zebrafish cerebellum. Zebrafish Purkinje cells have short axons (~8 µm; *Matsui et al., 2014*) that project locally to eurydendroid cells, which are analogous to neurons of the mammalian cerebellar nuclei. Thus, the medially located MCS cells described here are likely to target primarily medial eurydendroid cells, while SCS cells likely contact more lateral eurydendroid cells. Axons of medial and lateral eurydendroid cells form zebrafish cerebellar output pathways and project to different parts of the brain. Among the targets of the medial or rostromedial cerebellum are the thalamus, rostral optic tectum, and red nucleus, while the lateral cerebellum projects to the caudal tectal neuropil; additional gradients are evident in the rostro-caudal axis (*Heap et al., 2013*; *Matsui et al., 2014*). Purkinje cells in the medial and lateral cerebellum also contribute differentially to tail and eye movements evoked by optic flow (*Matsui et al., 2014*). Thus, the mediolateral gradient of MCS, ZCS, and SCS cells may indicate that different learned signals are sent to distinct brain regions, including those responsible for the initiation, maintenance, speed, and termination of swimming (*Brocard and Dubuc, 2003*; *Lambert et al., 2004*; *Soffe et al., 2009*; *Arrenberg et al., 2009*;

*Smetana et al., 2010*; *Kimura et al., 2013*; *Severi et al., 2014*; *Wang and McLean, 2014*; *Naumann et al., 2016*; *Juvin et al., 2016*).

## Relating naïve and learned responses of Purkinje cells

Given the variable nature of Purkinje cell responses during learning, a key question is whether pre-training synaptic responses can predict Purkinje activity that emerges during learning. In mammals, the absence of complex spikes to the US is predicted to limit long-term depression and possibly favor potentiation of parallel fiber responses (*Jörntell and Hansel, 2006*). Such plasticity rules appear consistent with those seen here in ZCS cells, which had the lowest likelihood of firing complex spikes to the US and developed greater parallel fiber responses during learning. Conversely, mammalian Purkinje cells that reliably generate complex spikes to a US are more likely to develop complex spike responses to a conditional stimulus (*Ohmae and Medina, 2015*). A similar result was seen here for MCS cells, which had the highest probability of firing complex spikes to the US. These cells also tended to produce multiple complex spikes during spontaneous swimming, suggesting that MCS cell-innervating climbing fibers are readily activated during motor commands. One possibility is that the responses of MCS cells resulted from innervation by multiple climbing fibers that had not yet been pruned over development (*Hsieh et al., 2014*), although the number of afferents could not be distinguished by the present experiments. Purkinje cells in each hemisphere differentiate in clusters, however, with a medial cluster shortly preceding a lateral cluster (*Hamling et al., 2015*), suggesting that the MCS cells described here may have been born earlier than ZCS and SCS cells, and might have had more rather than less time for pruning. More importantly, the pattern of changes in MCS cells, in which complex spikes became associated with the sensory rather than the motor response, is qualitatively different from SCS cells. The response of MCS cells therefore seems more likely to arise from distinct innervation patterns than solely from the degree of climbing fiber pruning.

## Complex spike plasticity in Purkinje cells

Much work has supported the idea that complex spikes function as instructive signals in learning, altering the production of simple spikes, often through heterosynaptic depression of parallel fiber inputs (*Gilbert and Thach, 1977*; *Mauk et al., 1986*; *Raymond et al., 1996*; *Medina et al., 2002*; *Guo et al., 2014*; *Yang and Lisberger, 2013*; *Ito et al., 2014*). In addition, complex spiking can itself be plastic. For instance, a learning-related reduction in complex spikes to the US has also been seen in rabbits and ferrets (*Sears and Steinmetz, 1991*; *Hesslow and Ivarsson, 1996*). Similarly, the MCS cells described here decreased their production of complex spikes during URs. A likely mechanism for such changes in mammals is an increased inhibition of inferior olivary neurons by cerebellar nucleo-olivary cells (*Medina et al., 2002*). Although homologous inhibitory output neurons in zebrafish have not yet been identified, the present data are suggestive of a similar pathway.

In contrast to their reduced response to the US, MCS cells increased their complex spiking to the CS. Similarly, in Purkinje cells of the eyeblink microzone in mice, complex spikes during the CS become more likely after training (*Ohmae and Medina, 2015*); conversely, in C3 Purkinje cells in decerebrate ferrets, complex spikes during the CS decrease over the course of training (*Rasmussen et al., 2014*). Monitoring the responses of MCS cells throughout training revealed that the increase in complex spiking developed even before learned swimming episodes occurred. Possible mechanisms for this plasticity within the afferent climbing fibers include potentiation of excitatory input that relays the CS to the inferior olive, modulation of intrinsic electrical properties of olivary neurons, and/or decreased CS-evoked inhibition from nucleo-olivary-like neurons. Indeed, recent studies provide evidence for modulation and plasticity of responses in the inferior olive (*Mathy et al., 2009*, *2014*; *Lefler et al., 2014*).

## A changing role for simple spikes during cerebellar learning

Mammalian Purkinje cells have high intrinsic simple spike rates (~50 spikes/s, *Thach, 1968*; *Häusser and Clark, 1997*), which can decrease during motor learning (*Gilbert and Thach, 1977*; *Hirata and Highstein, 2001*; *Blazquez et al., 2003*; *Jirenhed et al., 2007*; *Halverson et al., 2015*); lower simple spike rates presumably reduce the net inhibition of cerebellar output neurons and permit learned movements to occur. In zebrafish Purkinje cells, although simple spike rates are low (~6

Hz in the present data, ~10 Hz in *Scalise et al., 2016*), the experiments with Arch-activation illustrate a role for simple spiking that changes during training, depending on how reliably fish perform learned movements. In naïve fish, simple spike suppression impairs learning. In fish that have just begun expressing learned movements, suppression produces an incomplete deficit in CR expression. In fish that perform learned swimming with moderate reliability, suppression does not affect CR expression, but restoring simple spikes impairs performance. Only in fish that perform the CR with high reliability do learned responses continue without simple spikes and remain unaffected by simple spike restoration.

These results support the conclusion that, in larval zebrafish, simple spikes influence learning before and immediately after expression of the CR, but that plasticity supporting the execution of learned movements must also take place elsewhere. For example, Purkinje cell simple spikes could promote potentiation of excitatory synapses onto cerebellar output neurons. The idea that Purkinje-mediated inhibition instructs plasticity at target neurons that drive movements has been supported by modeling and experiments in mammals, both for the vestibulo-ocular reflex and for delay eyelid conditioning (*Miles and Lisberger, 1981*; *Medina and Mauk, 1999*; *Nguyen-Vu et al., 2013*; *Yang and Lisberger, 2014*). Synaptic plasticity under the control of Purkinje-mediated inhibition has also been demonstrated in vitro (*Aizenman et al., 1998*; *Pugh and Raman, 2006*; *McElvain et al., 2010*; *Pugh and Raman, 2008*; *Person and Raman, 2010*). In fact, plasticity within the zebrafish cerebellar circuit apparently adapted to a reduced level of simple spiking, since restoring simple spikes after suppression trials could decrease the likelihood of previously acquired CRs; in the simplest interpretation, the re-introduction of simple spiking effectively inhibited eurydendroid cells that helped drive the first CRs. This scenario could result from submaximal potentiation of excitation of cerebellar output neurons, so that restoration of Purkinje-mediated inhibition brought their responses to the CS below threshold. Thus, in the associative learning task explored here, learning is not all-or-none. Instead, plasticity in the cerebellar circuit continues after learned swimming emerges, and only appears to stabilize after fish reliably perform learned movements.

## Materials and methods

### Zebrafish

All procedures conformed to NIH guidelines and were approved by the Northwestern University Institutional Animal Care and Use Committee, protocol IS00000242 (IMR). Wildtype zebrafish (*Danio rerio*) were obtained from an in-house facility (Aquatic Habitats, Beverly, MA). Tg(Arch-tagRFP-T: car8:GCamp5) fish ('Arch[+]' fish), were kindly provided by Dr. Reinhard Köster (Technical University Braunschweig, Germany; *Matsui et al., 2014*) and were screened for RFP fluorescence at 5 days post-fertilization (dpf). Zebrafish were housed in system water (28.5°C, pH = 7.3, conductivity = 550 μS) and maintained on a 14 hr light:10 hr dark cycle. Experiments were done during the light phase (between 10 am and 7 pm), at room temperature (~22°C), on larval fish (6–8 dpf, before sexual differentiation). The time of day of the experiment did not differ between learner and non-learner fish (p=0.32, unpaired t-test), so data from all times of day were pooled. MCS, SCS and ZCS cells were recorded from fish of comparable ages (6.5 ± 0.2, 6.4 ± 0.2, and 6.5 ± 0.2 dpf, respectively, $F_{(2,30)}$=0.03, p=0.98).

### Electrophysiological recordings

Recordings were performed based on those described previously (*Drapeau et al., 1999*; *Masino and Fetcho, 2005*; *Wang and McLean, 2014*). Each fish was immobilized by 3 min immersion in α-bungarotoxin (1 mg/ml, Tocris, Bristol, United Kingdom) in system water followed by 5 min in extracellular solution containing (mM): 134 NaCl, 2.9 KCl, 2 MgCl$_2$, 10 HEPES, 10 glucose, and 2.1 CaCl$_2$, buffered to pH 7.8 with NaOH, with final osmolarity 290 mOsm. The immobilized fish was transferred to a Sylgard-lined plastic recording chamber containing extracellular solution, with 0.01% MS-222 anesthetic added for experiments involving dissection for neuronal recordings or ablation. Blood circulation was monitored throughout the dissection. The fish was secured to the Sylgard surface with pins so the dorsal side of the head and the left side of the tail faced up. A midline incision was made and the skin was pinned to expose the brain. For recordings from peripheral motor nerves (ventral roots) along the tail, the skin was removed on the left side, from the rostral edge of the

swim bladder to 3–5 segments rostral to the tail tip. For ablation experiments, the cerebellum was scored manually with a tungsten needle and aspirated through a 50–60 μm diameter micropipette positioned with a micromanipulator (MP-385, Sutter, Novato, California). Fish with damage to the tectum or brainstem or with impaired circulation were not used for experiments. After dissection, solution was exchanged for saline without MS-222 for recordings.

The brain was visualized with IR-DIC microscopy on a FS2 Axioskop (Zeiss, Oberkochen, Germany). The locations of electrophysiologically characterized Purkinje cells were captured with a SensiCam camera (PCO.Imaging, Kelheim, Germany) and/or from the coordinates of the Sutter MP-385 manipulator. Borosilicate patch pipettes were pulled to tip resistances of 8–12 MΩ and filled with intracellular solution containing (mM): 120 K-gluconate, 12 Na-gluconate, 3.2 NaCl, 2 MgCl$_2$, 0.025 CaCl$_2$, 1 EGTA, 0.3 mM Tris-GTP, 1 MgATP, 14 creatine phosphate, 10 HEPES, and 3 Alexa Fluor 488 hydroxide, buffered to pH to 7.4 with KOH. Whole-cell recordings were made with a Multiclamp 700B and Digidata 1322A with pClamp software (Molecular Devices, Sunnyvale, California) from Purkinje neurons in the left hemisphere of the cerebellum. Data were acquired at 50 kHz and filtered at 10 kHz. All voltage clamp recordings were made at a holding potential of −60 mV. In current clamp, the amplifier injected a constant 20 pA current even in 'I = 0' mode, which artificially depolarized cells in initial experiments. The depolarization could reduce the amplitude of complex spikes and inactivate voltage-gated currents required for full amplitude simple spikes, which was taken into account in the analysis. In all subsequent current-clamp recordings, an equivalent hyperpolarizing current was applied to neutralize this current. Bridge balance and capacitance neutralization were also applied.

For ventral root recordings, patch pipettes were cut to a 20–50 μm tip diameter, heat-polished, and bent to ~20° to improve contact with the body wall. The pipette was filled with extracellular solution and placed on the intermyotomal cleft. Recordings from the ventral root were made with an Axopatch 200B (Molecular Devices) amplifier in current clamp mode with low and high frequency cutoffs of 300 and 4000 Hz.

## Behavior

Presentation of sensory stimuli was controlled with a Master-8 Pulse Stimulator (A.M.P.I, Jerusalem, Israel) triggered by pClamp. The conditional stimulus (CS) was a 2 s light (unless otherwise stated) from a blue LED (470 nm, 53 lux) surrounded by a 3 cm aluminum foil disk, positioned ~5 cm above and ~30° to the right of the fish's head. The unconditional stimulus (US) was a brief electrical current (1 mA, 5 ms) applied to the tail tip by a concentric bipolar stimulating electrode (WPI, Sarasota, Florida), which mimics a mild tactile stimulus (*Kahn and Roberts, 1982*). Experiments were conducted with the microscope light on (1100–4500 lux) and the objective placed directly above the fish. The environment was homogeneous with no visual cues in close proximity to the fish, which minimized the possibility of behaviors driven by a visuomotor mismatch during fictive swimming (*Patterson et al., 2013*).

An initial test US was applied to verify positioning of electrodes for ventral root recording and tactile stimulation. Next, a test CS was applied, and if the light evoked fictive swimming, the microscope light luminance was increased to reduce contrast with the CS light until the CS no longer elicited swimming. Six fish were discarded for persistent light-evoked or excessive spontaneous swimming. For experiments including Purkinje cell recordings, the CS and US were each presented alone first, once in current clamp and once in voltage clamp. Training consisted of paired presentations of the CS followed immediately by the US. Trials were triggered manually, with start-to-start inter-trial interval of 40–55 s, for a maximum of 70 trials or the duration of the Purkinje cell recording (15–45 trials). Pseudoconditioning consisted of equal numbers of CS and US presentations given randomly at intervals of 15–25 s.

Archaerhodopsin-3 (Arch) was activated by a green LED (565 nm; 5400 lux; Thor Labs, Newton, New Jersey). The size of the illuminated area was adjusted with an iris diaphragm. To activate Arch, green light was directed at the whole cerebellum. For control trials with illumination displaced, the green light was directed ~1 mm rostral to the head. In Arch$^+$ Purkinje cells, no Arch-dependent currents were evoked either by displaced illumination or by the blue CS light (N = 6). Outcrossing Tg (Arch-tagRFP-T:car8:GCamp5) fish produced equal ratios of Arch$^+$ and Arch$^-$ larvae. Either wild-type or Arch$^-$ fish were used in control and pseudoconditioned groups and were counterbalanced for age.

## Analysis of electrophysiological events

Electrophysiological data were analyzed in IGOR-Pro (Wavemetrics, Lake Oswego, Oregon). The electrophysiological recordings from Purkinje cells were differentiated and dV/dt peaks were used to find putative synaptic and/or action potentials. After the identification procedure described below, every event was checked and corrected if necessary by assessing the underlying conductance extracted from the dV/dt, as described and illustrated in the *Results.*

The procedure was first optimized to identify complex spikes. We determined a reasonable starting threshold as follows: Voltage clamp recordings of climbing fiber EPSCs, which were reliably evoked by the tactile stimulus, had a mean amplitude of $-280 \pm 29$ pA at $-60$ mV (N = 27 EPSCs in 27 cells), and large spontaneously occurring putative climbing fiber EPSCs were $-253 \pm 29$ pA (N = 222 EPSCs in 39 cells). For Purkinje cells, which have a mean capacitance of $10.7 \pm 0.67$ pF (N = 29 cells), a current of $-250$ pA would give a dV/dt of 25 mV/ms. In current clamp, however, cells were often more depolarized, owing to synaptic potentials (and extrinsic current from the amplifier in initial experiments, described above), such that complex spikes often arose from voltages above $-60$ mV. Therefore, to identify complex spikes in each differentiated record, threshold was first set at 20 mV/ms. The extracted events were then examined by eye (in the raw and the differentiated records) to assess whether large, rapid depolarizing events arising directly from the baseline were adequately selected; the membrane voltage just preceding the event was considered in this evaluation. If necessary, the threshold would be adjusted to a lower value. After events were detected with an appropriate threshold, the record was re-examined by eye to eliminate false positives, which were rare but could include secondary humps on either complex or simple spikes.

After complex spikes were extracted from the record, threshold was lowered to account for the next-most rapidly depolarizing events, which were identified as putative simple spikes. These were evident only when the membrane potential was sufficiently hyperpolarized for the recruitment of voltage-gated channels. The threshold was initially set at 8 mV/ms. After events were detected, the raw and differentiated records were once more inspected by eye to eliminate false positives, such as secondary humps on simple or complex spikes and to verify that detected events resembled action potentials with a threshold inflection. On average the dV/dt of these events was $10.5 \pm 0.6$ mV/ms (N = 46 cells), corresponding to a current of about $-114$ pA (for a ~10 pF cell). Since parallel fiber EPSCs are small (<20 pA at $-60$ mV, i.e.,<333 pS), voltage-gated Na currents likely predominated in setting this rate of rise. This factor was taken into account when the underlying conductances were calculated (see *Results*).

After extraction of simple spike events, the dV/dt threshold was lowered again to identify pfEPSPs. This value was set at 0.4 mV/ms. This corresponds to an amplitude of 40 pA, more than twice the amplitude of EPSCs visible in voltage clamp records. This threshold was then reduced by eye to capture voltage deflections with a pfEPSP-like waveform that were distinct from high-frequency fluctuations (baseline noise) throughout the trace. In addition, these events were constrained to occur at voltages that were below the threshold of simple spikes. After events were detected by thresholding, the record was once more inspected by eye to eliminate false positives, including subthreshold depolarizations that did not become full-fledged simple spikes that had already been counted as a pfEPSP. These criteria were relatively stringent and were intentionally selected to give incorrect rejection errors (small pfEPSPs or pfEPSPs occurring during periods of simple spikes) rather than false positives. In cells that showed pfEPSPs during swimming events, synaptic activity was often elevated for time scales that were commensurate with the duration of swimming, i.e., a few hundred ms. Therefore, for estimation of latencies of pfEPSPs relative to swimming, measurements were restricted to pfEPSPs that initiated depolarizations that lasted at least 200 ms.

During behavioral tasks, hyperpolarizations were identified as deflections below the most negative voltage during the pre-trial baseline. For ventral root signals, voltage recordings were rectified, smoothed by a moving average in a 2 ms window, and thresholded to detect individual bursts. Fluorescent images were obtained on a confocal microscope (Zeiss LSM 710, 40x) with excitation wavelength 488 nm, and analyzed with Zen imaging software.

## Statistical analysis

N values for cells and fish (biological replicates) are given in the text. Data are reported as mean ± SEM, unless noted. Statistics were calculated with Microsoft Excel and SPSS (IBM), with tests as

follows: Two-sided paired or unpaired t-tests, for comparisons of two groups or two measures from one group (*Figures 5H* and *6D* [one paired CS and CR CV per cell], *8E* and *9A,D*); One-way ANOVA with Bonferroni *post hoc* corrections as needed, for groups of fish and/or cells (*Figures 2F, H*, *3B*, *4F*, *5C* and *7D*); repeated measures ANOVAs with Tukey *post hoc* corrections as needed, for comparisons across training (*Figures 3C, E*, *5E, F, G*, *7C* and *8D* left); chi-squared tests, for the proportion of learners among Arch[+] and control fish (*Figure 8D* right). For data reported only in the text, the statistical test is indicated. F-statistics and chi-square statistics are given in legends.

## Acknowledgements

We are grateful to Professor Reinhard Köster (Technical University Braunschweig, Germany) for the Tg(Arch-tagRFP-T:car8:GCamp5) zebrafish; Matt Chiarelli and Elissa Szuter for technical help maintaining the fish colony; and members of the Raman lab for comments on an earlier version of the manuscript.

Supported by NIH R37-NS39395 (IMR), a Brain Research Foundation Pilot Grant (IMR), NIH F31-NS095476 (TCH), NIH T32-MH067564 (trainee, TCH), and NIH R01-NS067299 (DLM). Funding sources were not involved in study design, data collection or interpretation, or decision to submit the work for publication.

## Additional information

### Competing interests

IMR: Reviewing editor, *eLife*. The other authors declare that no competing interests exist.

### Funding

| Funder | Grant reference number | Author |
| --- | --- | --- |
| National Institutes of Health | R37-NS39395 | Indira M Raman |
| Brain Research Foundation | Pilot Grant | Indira M Raman |
| National Institutes of Health | F31-NS095476 | Thomas C Harmon |
| National Institutes of Health | T32-MH067564 | Thomas C Harmon |
| National Institutes of Health | R01-NS067299 | David L McLean |

The funders had no role in study design, data collection and interpretation, or the decision to submit the work for publication.

### Author contributions

TCH, Conceptualization, Formal analysis, Funding acquisition, Validation, Investigation, Visualization, Writing—original draft, Writing—review and editing; UM, Formal analysis, Investigation; DLM, Conceptualization, Formal analysis, Supervision, Funding acquisition, Validation, Visualization, Writing—review and editing; IMR, Conceptualization, Formal analysis, Supervision, Funding acquisition, Validation, Visualization, Writing—original draft, Project administration, Writing—review and editing

### Author ORCIDs

Indira M Raman, http://orcid.org/0000-0001-5245-8177

### Ethics

Animal experimentation: All experiments were done in accordance with federal and institutional guidelines and were approved by the Northwestern University IACUC, protocol IS0000242 (IMR), Animal Welfare Assurance number A3283-01.

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
