## [Decision Letter]

Thank you for submitting your article "Distinct contributions of multiple classes of Purkinje neurons to associative motor learning in larval zebrafish" for consideration by *eLife*. Your article has been reviewed by three peer reviewers, and the evaluation has been overseen by a Reviewing Editor and Eve Marder as the Senior Editor. We apologize for the long time required to arrive at this decision, but there were an inordinate number of delays at multiple steps in the editorial process, from delays in securing reviewers, obtaining their reviews and synthesizing the decision. The reviewers have opted to remain anonymous.

The reviewers have discussed the reviews with one another and the Reviewing Editor has drafted this decision to help you prepare a revised submission.

Summary:

This study addresses an important topic in cerebellar physiology: how does the diversity of Purkinje cells contribute to cerebellar function and learning? The authors take full advantage of the power of the zebrafish preparation and the electrophysiological data are of very high quality. The paper has two significant components. The first is distinguishing three potential functional classes of Purkinje cells based on their spiking properties, namely the number of complex spikes fired during learning. These assignments are bolstered by the finding that cells that differ functionally tend to be located in different lateral-medial positions in the cerebellum.

What is not clear is how these different functional classes contribute to learning and behavior, which is well beyond the scope of this paper. (Because of this, the authors might consider changing the title of the paper.) The final component of the paper uses the inhibitory opsin Arch to suppress simple spiking during different phases of the learning paradigm. These elegant experiments indicate that the functional role of simple spiking changes as learning proceeds, being essential for learning the task initially, but not necessary for performing the task after learning has solidified. In some ways, this is the more interesting part of the paper because of its more mechanistic focus. Given long-standing controversies about the roles of Purkinje cell activity in cerebellar learning, this is an important set of findings. However a potentially major caveat is that the immaturity of the larval zebrafish cerebellum, the extremely low firing rates of complex and simple spikes, and the poorly timed nature of the conditioned responses make it difficult to know whether the results can be more broadly generalized to the mature mammalian cerebellum.

Essential revisions:

1) The paper is densely written and difficult to read. The figure legends contain numerous mistakes. There is a fair amount of jargon. Terms like maintenance and expression (of learning) are used but not clearly defined. The paper should be carefully edited to improve clarity and eliminate inconsistencies in the figures and legends.

2) The paper provides relatively few details about the experimental procedures. It is known that zebrafish activity varies during the day. The authors do not address whether they took this into account in designing the experiments. According to the methods section, up to 70 trials of the blue light followed by the tactile stimulation were applied. Among learners, episodes of fictive swimming during the light pulse increased in frequency and decreased in latency with repeated trials. To put these results into a broader context of learning and behavior, it would have been interesting to know whether the training paradigm affects the behavior of fish freed from immobilization after the training. Do such fish increase their swimming or initiate swimming in response to blue light applied?

3) The authors give criteria for distinguishing complex and simple spikes, and cf and pf EPSPs using measurements of dV/dt and amplitude. In addition to the relative amplitudes of cxs and ss shown in Figure 1, it would be useful to show the bulk data (or provide source files) of dV/dt and amplitude values so that the reader can evaluate whether they are clearly bimodal and the extent to which the distributions might overlap. The figure should be labeled to indicate the criteria used to categorize the events. Given the small amplitude of the pf EPSPs, any criteria used to distinguish actual events from random noise should be provided.

4) The records shown in Figure 2 are short. In several of the cells, the interval prior to swimming is devoid of spontaneous activity, which presumably reflects the relatively low frequency of simple spiking. In many cases, swimming is accompanied by a dramatic increase in firing frequency that dramatically outlasts swimming behavior. It would be interesting to see the bulk data (or provide source files), which could be presented as raster plots of firing on a longer time scale before, during and after swimming, with the interval of swimming indicated by symbol color or some other means. Do firing properties differ when spontaneous and evoked swimming are compared? This might be the case, for instance, if the tactile stimulus activates the escape response rather than simple swimming. Admittedly, this is a bit off the point of Purkinje cell diversity, but the data would likely be of interest to other researchers in the field.

5) There is a disconnect between complex spike-based functional characterization and optogenetic simple spike suppression. The authors characterize three types of Purkinje cells depending on the pattern of complex spike responses to the learned behavior, and find clear differences in their properties developed with learning as shown in Figure 5 and Figure 6. However, the role of these complex spikes in learning is largely ignored in the optogenetic experiments and in the Discussion. The authors should at least show the properties of complex spike firing in a representative zebrafish during the task with Arch activation. Or they could also address the differential role of complex spike in learning by olivary lesions. Vice versa, there seems significant modulation of simple spike firing around the CR in SCS cells as shown in Figure 4, Figure 6-B. This should also be fully analyzed like complex spikes, and simple spike-based classification should also be considered.

6) The validation of spike classification and EPSP detection could be improved. Zebrafish Purkinje neurons seem to lack characteristic waveforms such as spikelets for complex spikes compared with simple spikes, unlike in mammals. The determinant of the former is only the amplitude and rise time. For example, in Figure 5, the fourth complex spike in middle trace and a simple spike between third and fourth complex spikes in late trace are pretty much similar. As the timing of each firing pattern is critical for the interpretation of the data, the authors could validate this classification of firing pattern by artificial climbing fiber stimulation. In addition, detection of EPSP is not clear because there is only one trace (Figure 1) to show parallel fiber inputs which are barely above noise fluctuation. I would like to see representative EPSP traces of parallel fiber inputs for figures such as Figure 2/H and Figure 7 that show how they calculate the latency and count the number of them which are summated to generate simple spike.

7) The authors use early (first 3 trials), middle (3 trials before CR acquisition) and late (3 trials after CR acquisition) as a measure of learning in the analysis of MCS, SCS and ZCS cells. However for the optogenetic part, 1, 3 or 6-CR (1, 3 or 6 consecutive CR) are used for the measure. These double standards are confusing and make it difficult to assign the former complex spike data to the latter optogenetic results.

8) The authors should quantify how their cerebellar ablation and Arch illumination affects baseline swimming behaviour. A general alteration of swimming frequency and/or amplitude would be a major confound to the interpretation of their data.

9) The interpretation of the different response types in Figure 2 is a bit tenuous. In the experiments done in the fictive, paralyzed preparation, the authors should make it clear whether there were any visual cues present during the experiment. Specifically, it seems that the complex spikes observed in spontaneous and sensory-evoke swimming may be due to a visuomotor mismatch if there was sufficient illumination present to give the fish landmarks that would be expected to translate as it swam. An alternative hypothesis to the conclusions at the end of paragraph five of the Results section: The tactile stimulus results in a reflexive swim locally in the tail means that the fish swims immediately (this is evident in the latency to swim when comparing visual and tactile stimuli). The 50 ms lag in response is the amount of time it takes this sensory stimulus to be processed. Indeed, Figure 2 shows that the pf EPSP latency for visual and tactile stimuli is approximately the same with respect to stimulus onset.

10) In Figure 4, the authors assert that the distinction between MCS and SCS cells cannot be explained by differences in the number of CFs innervating each cell (i.e. incomplete elimination). To solidify this claim, the authors should analyse the consistency of sizes of their complex spikes in response to the conditioned stimulus. Specifically, it would be helpful to describe whether and how the different climbing fiber responses are manifested during the learning process – is one of the inputs preferentially active? Is the same climbing fiber the active one during each of the multiple complex spikes within a trial (for MCS only) and is the same climbing fiber activated across trials (for MCS and SCS responses)?

11) The Introduction and first paragraph of Discussion focuses on the question of whether Purkinje cells comprise functionally distinct classes, but it has long been appreciated that Purkinje cells in different regions of the mature cerebellum are functionally distinct (with respect to physiological response properties, anatomical circuit connections, and roles in very different types of behavior). We already know (e.g. from publications on the oculomotor and vestibular cerebellum) that "simple and complex spikes signal different sensory or motor components associated with a behavioral task" (raised as a question in the Introduction). In light of anatomical work on zebrafish cerebellum, it would be surprising if Purkinje cells in this species were homogenous in their firing properties. In the Discussion section, first paragraph, it would be appropriate to insert "in fish as in other vertebrates" into the topic sentence "the present results demonstrate, however, that not all Purkinje cells respond identically to sensory or motor signals". The Discussion paragraph starting with "Evidence for different classes of Purkinje cells has also been found in the mammalian cerebellum" similarly ignores decades of physiological analyses of Purkinje cells that demonstrate differences in response properties. I suggest that the term "class" be defined more precisely and that the modular organization of inferior olivary projections to the cerebellum be explicitly discussed as it pertains to fish, birds, and mammals.

12) There is a puzzling disconnect between the known anatomical projections of zebrafish Purkinje cells and the different classes of Purkinje cells reported in this paper. How do the different firing patterns evoked during visual stimuli, somatosensory stimuli, and fictive swimming relate to the differential projections of medial vs lateral Purkinje cells. A new paper from Sawtell's group describing activity of presumably the same Purkinje cells as those recorded for this study is relevant for this issue (https://www.ncbi.nlm.nih.gov/pubmed/27512018).

13) The associative conditioning paradigm and resulting behavioral (nerve root activity) output are a bit puzzling and merit more discussion. The light is on for 2 seconds prior to the tail shock – this is a very long time for the cerebellum and seems particularly long in the short life of a baby zebrafish. What motivated this paradigm? Did a shorter visual stimulus (e.g. 500 ms or 1s) work as well? Does the apparently stochastic timing of the conditioned response reflect immaturity of the larval zebrafish cerebellum, or something about the prolonged timing of the visual stimulus? If they are conditioned responses, do they extinguish after repeated visual stimuli in the absence of somatosensory stimulation?

14) It would be very helpful to discuss what might be learned of general relevance for cerebellar function from studies of the larval zebrafish cerebellum in which firing rates are an order of magnitude lower than in mature cerebella from other vertebrates. Certainly this (and the recent Sawtell paper) raise intriguing questions about cerebellar evolution and whether there are fundamental computations that work robustly in both tiny sluggish cerebella and large speedy cerebella. How much do the conclusions depend on developmental stages? Please discuss the implications of multiply innervation by climbing fibers and low firing rates for the interesting and potentially general conclusions of this study about the dynamic changes in Purkinje cell spiking and functional role of simple spikes during associative conditioning.

---

## [Author Response]

Essential revisions:

1) The paper is densely written and difficult to read. The figure legends contain numerous mistakes. There is a fair amount of jargon. Terms like maintenance and expression (of learning) are used but not clearly defined. The paper should be carefully edited to improve clarity and eliminate inconsistencies in the figures and legends.

We have extensively edited the manuscript throughout to organize the ideas more explicitly, to define terminology, to simplify the text, and to clarify illustrations. Definitions are included of acquiring CRs (subsection “A cerebellar learning task in the zebrafish”) and of acquisition, expression, and maintenance of learning (subsection “Suppressing Purkinje cell simple spikes”). We regret the errors in the legend of Figure 1, which arose when the figure was reorganized just before submission. We have corrected the mismatches.

2) The paper provides relatively few details about the experimental procedures. It is known that zebrafish activity varies during the day. The authors do not address whether they took this into account in designing the experiments. According to the methods section, up to 70 trials of the blue light followed by the tactile stimulation were applied. Among learners, episodes of fictive swimming during the light pulse increased in frequency and decreased in latency with repeated trials. To put these results into a broader context of learning and behavior, it would have been interesting to know whether the training paradigm affects the behavior of fish freed from immobilization after the training. Do such fish increase their swimming or initiate swimming in response to blue light applied?

Information on time-of-day of experiments was in the original manuscript, but additional analyses of whether the time of experiments affected the experimental outcomes is now included in the Materials and methods, first paragraph. The time of day of testing did not differ between learning and non-learning fish.

Regarding the reviewer’s request to free the fish, the fish cannot be freed from immobilization after training because α-bungarotoxin is irreversible; survival on the time scale of receptor turnover (~24 hours) is likewise unfeasible and inappropriate because fish are pinned and the skin and other tissue over the brain and muscle have been removed, so survival is unlikely. Since ventral root activity indeed occurs in response to the conditioning light, it follows that a freely moving fish would indeed swim after training, at least until extinction occurs (in the absence of reinforcement as tactile stimulus to the tail).

Other details on experimental procedures have been added. Please also see point 3.

3) The authors give criteria for distinguishing complex and simple spikes, and cf and pf EPSPs using measurements of dV/dt and amplitude. In addition to the relative amplitudes of cxs and ss shown in Figure 1, it would be useful to show the bulk data (or provide source files) of dV/dt and amplitude values so that the reader can evaluate whether they are clearly bimodal and the extent to which the distributions might overlap. The figure should be labeled to indicate the criteria used to categorize the events. Given the small amplitude of the pf EPSPs, any criteria used to distinguish actual events from random noise should be provided.

The explanation was admittedly too cursory. We now provide a complete explanation of how events were distinguished (Materials and methods subsection “Analysis of electrophysiological events”; Results subsection “Identification of events”). We also validated the approach by analyzing the conductances underlying each event (as a measure that is independent of membrane potential, unlike dV/dt and amplitude), and the distribution of the conductances of all events included in the study is illustrated in Figure 1.

4) The records shown in Figure 2 are short. In several of the cells, the interval prior to swimming is devoid of spontaneous activity, which presumably reflects the relatively low frequency of simple spiking. In many cases, swimming is accompanied by a dramatic increase in firing frequency that dramatically outlasts swimming behavior. It would be interesting to see the bulk data (or provide source files), which could be presented as raster plots of firing on a longer time scale before, during and after swimming, with the interval of swimming indicated by symbol color or some other means. Do firing properties differ when spontaneous and evoked swimming are compared? This might be the case, for instance, if the tactile stimulus activates the escape response rather than simple swimming. Admittedly, this is a bit off the point of Purkinje cell diversity, but the data would likely be of interest to other researchers in the field.

Longer records (>3 sec) are shown in Figure 1, Figure 5, Figure 6, and Figure 7, which include spontaneous firing by Purkinje cells as well as spontaneous swimming recorded at the ventral root. The records are short in Figure 2 to illustrate the variety of responses to swimming with a time base that permits the spikes to be resolved. The point of Figure 2 is that there was substantial variance across cells; the later experiments help us draw the conclusion that a fraction of this variance may relate to the existence of different cell populations. The text has also been clarified to help readers follow the point of the figure.

Regarding the question of Purkinje cell spike responses to different kinds of swimming, we quantify in Figure 2 the latencies to climbing fiber-driven complex spikes and parallel-fiber driven EPSPs/simple spikes for spontaneous, visual and tactile-evoked swimming. Because of the different populations of Purkinje cells that the study identifies, it is not instructive to make further comparisons of firing during spontaneous and evoked swimming by pooling Purkinje populations. Comparisons of spike number and/or latency for spontaneous, learned, and unconditional response-related swimming are indeed given, however, in cases when the three types of swimming reliably evoked one type of response, i.e., for complex spikes for MCS cells and SCS cells (Figure 5, Figure 6) and for pfEPSPs for ZCS cells (Figure 7).

Additionally, the responses of all cells followed through learning are included in Figure 4. The time course of pfEPSP/simple spikes and hyperpolarizations has been added (grey bars) to show their relation to the onset of learned swimming (t=0). Complex spike probability with respect to the US is also reported for each cell group (subsection “Purkinje cell activity during learning”).

As the reviewer notes, a further general analysis of firing properties is off the point and the manuscript is already quite complex with three separate components to it. We therefore think that further exploration of this topic is something to be addressed in a separate study.

5) There is a disconnect between complex spike-based functional characterization and optogenetic simple spike suppression. The authors characterize three types of Purkinje cells depending on the pattern of complex spike responses to the learned behavior, and find clear differences in their properties developed with learning as shown in Figure 5 and Figure 6. However, the role of these complex spikes in learning is largely ignored in the optogenetic experiments and in the Discussion. The authors should at least show the properties of complex spike firing in a representative zebrafish during the task with Arch activation. Or they could also address the differential role of complex spike in learning by olivary lesions. Vice versa, there seems significant modulation of simple spike firing around the CR in SCS cells as shown in Figure 4, Figure 6. This should also be fully analyzed like complex spikes, and simple spike-based classification should also be considered.

We have added data to Figure 8, and panel Figure 8 now shows a sample Purkinje cell (MCS) through learning and during Arch activation. It illustrates the suppression of simple spikes but not complex spikes during learned swimming. The associated text is in paragraph three of subsection “Suppressing Purkinje cell simple spikes”.

Regarding the disconnect between classifying cells by complex spikes and Arch’s ability to suppress simple spikes, we did not know *a priori* whether hyperpolarization by Arch activation would block complex spikes as well as simple spikes. The plan was to develop a learning task, record Purkinje activity, and then manipulate Purkinje activity through the optogenetic means available. Because Arch blocked simple but not complex spikes, we used it to identify roles for simple spikes. We have tried to re-emphasize this point in the revision of the manuscript.

Likewise, the decision to categorize cells based on complex spike patterns was not an *a priori* decision but came from inspection of the data set. Simple spike-based classifications were indeed considered, but most appeared arbitrary. Classifying by 0, 1, or >1 complex spikes included all Purkinje cells, and the categorization could then be validated through additional measurements. We have added text to clarify the reasons for the classification used and to indicate that other classifications might be possible, subsection “Purkinje cell activity during learning”. Please note that ZCS cells showed parallel fiber activity rather than complex spikes associated with learned responses, so measures of parallel fiber activity are not absent from the study. To emphasize this point, the number of parallel-fiber driven EPSPs over the course of training in MCS and SCS cells has now been added to Figure 7 (subsection “Zero complex spiking cells”). Unlike in ZCS cells, they do not increase.

What the roles are of complex spikes of the different cell types is indeed an interesting question, but is likely to remain unanswered even by lesion of the inferior olive, which would likely abolish cerebellar learning and otherwise disrupt cerebellar physiology. Also, the present work provides evidence for multiple olivary neuron populations differentially innervating Purkinje cell populations. The organization of climbing fiber projections has yet to be described in larval zebrafish. Therefore, a lesion of the entire inferior olive would not clarify the role of complex spikes within MCS and SCS populations, nor would it provide information about ZCS cells. In short, while the present work provides the foundation and motivation for probing the role of complex spikes during training (and olivocerebellar projections), doing so extends beyond the scope of the present manuscript. Although the role of complex spikes in learning can only be inferred from our experiments, we address the possible role of complex spikes in MCS cells in the Discussion section entitled “Complex spike plasticity in Purkinje cells” by relating our findings to those from delay eyelid conditioning in mammals.

6) The validation of spike classification and EPSP detection could be improved. Zebrafish Purkinje neurons seem to lack characteristic waveforms such as spikelets for complex spikes compared with simple spikes, unlike in mammals. The determinant of the former is only the amplitude and rise time. For example, in Figure 5, the fourth complex spike in middle trace and a simple spike between third and fourth complex spikes in late trace are pretty much similar. As the timing of each firing pattern is critical for the interpretation of the data, the authors could validate this classification of firing pattern by artificial climbing fiber stimulation. In addition, detection of EPSP is not clear because there is only one trace (Figure 1) to show parallel fiber inputs which are barely above noise fluctuation. I would like to see representative EPSP traces of parallel fiber inputs for figures such as Figure 2 and Figure 7 that show how they calculate the latency and count the number of them which are summated to generate simple spike.

As indicated in the response to Point 3, a detailed description of how events were distinguished has been added to Methods and Results, and a complete histogram of events is now given in Figure 1. Instead of using artificial climbing fiber stimulation as suggested, we calibrated the measurements to cfEPSCs evoked by the US. The fourth complex spike was mislabeled on the figure but not miscounted. The tick mark has been removed. Traces of enlarged pfEPSPs/pfEPSCs have also been added to Figure 1 and Figure 7.

7) The authors use early (first 3 trials), middle (3 trials before CR acquisition) and late (3 trials after CR acquisition) as a measure of learning in the analysis of MCS, SCS and ZCS cells. However for the optogenetic part, 1, 3 or 6-CR (1, 3 or 6 consecutive CR) are used for the measure. These double standards are confusing and make it difficult to assign the former complex spike data to the latter optogenetic results.

This comment seems to be based on a misunderstanding. The approaches analyze two different experiments measuring two different things. In the first, we examined how response properties develop over the course of learning, so we looked at early trials (before learning), middle trials (just before performance of CRs) and late trials (after performance of CRs). This is a descriptive analysis illustrating attributes of unperturbed learning over time. In the second, we tested to what extent learning persisted after fish had attained different learning criteria. This is an experimental analysis revealing whether the response to simple spike suppression after learning is different when fish had learned to a 1-CR, 3-CR or 6-CR criterion, which it does. Because these two forms of analysis teach us two different things, we have retained them. The text in subsection “Multiple complex spike cells” where “early,” “middle,” and “late” trials are defined has been edited to avoid this misunderstanding.

8) The authors should quantify how their cerebellar ablation and Arch illumination affects baseline swimming behaviour. A general alteration of swimming frequency and/or amplitude would be a major confound to the interpretation of their data.

Swimming behavior during Arch activation does not change as was already described in the text (CR latency, swimming frequency, amplitude, subsection “Simple spike suppression following CR acquisition”).

Swimming frequency and amplitude are not relevant to interpreting how learning was affected by cerebellar ablation, so any changes in these parameters do not influence the results and interpretations. Ablation was used to test whether learned responses emerged, which they did not. Traces from cerebellum-ablated fish illustrating swimming responses to the US and to high-contrast visual stimulation are now shown in Figure 3 to help clarify this point.

9) The interpretation of the different response types in Figure 2 is a bit tenuous. In the experiments done in the fictive, paralyzed preparation, the authors should make it clear whether there were any visual cues present during the experiment. Specifically, it seems that the complex spikes observed in spontaneous and sensory-evoke swimming may be due to a visuomotor mismatch if there was sufficient illumination present to give the fish landmarks that would be expected to translate as it swam. An alternative hypothesis to the conclusions at the end of paragraph five of the Results section: The tactile stimulus results in a reflexive swim locally in the tail means that the fish swims immediately (this is evident in the latency to swim when comparing visual and tactile stimuli). The 50 ms lag in response is the amount of time it takes this sensory stimulus to be processed. Indeed, Figure 2 shows that the pf EPSP latency for visual and tactile stimuli is approximately the same with respect to stimulus onset.

This is an interesting idea but seems unlikely to account for the results, since no visual cues were present within range of the larval zebrafish’s nearfield vision (Patterson et al., 2013), now stated in the Materials and methods, subsection “Behavior*.”*. Also, tactile swimming is not reflexive through the spinal cord but requires the brainstem to generate high-speed swimming, as it is removed by spinal transection (Bhatt, McLean, Hale and Fetcho, 2007), now cited in the Results section. These points notwithstanding, if complex spikes to swimming indicated a mismatch, most swimming events (spontaneous, visual, tactile, and learned) would include complex spikes, which would always lag swimming onset, which was not the case. This point is now stated explicitly in the Results section. Nevertheless, we have rephrased the text to make clear the intent of its being simply a summary of the observations.

10) In Figure 4, the authors assert that the distinction between MCS and SCS cells cannot be explained by differences in the number of CFs innervating each cell (i.e. incomplete elimination). To solidify this claim, the authors should analyse the consistency of sizes of their complex spikes in response to the conditioned stimulus. Specifically, it would be helpful to describe whether and how the different climbing fiber responses are manifested during the learning process – is one of the inputs preferentially active? Is the same climbing fiber the active one during each of the multiple complex spikes within a trial (for MCS only) and is the same climbing fiber activated across trials (for MCS and SCS responses)?

We have removed the estimates of the number of climbing fibers innervating MCS and SCS cells. We did pursue a more in-depth analysis of EPSCs but concluded that there are too many confounding factors for unequivocal interpretation (namely, depression and the possibility of multiple climbing fibers of comparable strengths). We therefore mention multiple innervation in the discussion but emphasize the physiological rather than the mechanistic result, namely, that the patterns of firing by MCS and SCS cells are so different they probably arise from climbing fibers carrying different information.

11) The Introduction and first paragraph of Discussion focuses on the question of whether Purkinje cells comprise functionally distinct classes, but it has long been appreciated that Purkinje cells in different regions of the mature cerebellum are functionally distinct (with respect to physiological response properties, anatomical circuit connections, and roles in very different types of behavior). We already know (e.g. from publications on the oculomotor and vestibular cerebellum) that "simple and complex spikes signal different sensory or motor components associated with a behavioral task" (raised as a question in the Introduction). In light of anatomical work on zebrafish cerebellum, it would be surprising if Purkinje cells in this species were homogenous in their firing properties. In the Discussion section, first paragraph, it would be appropriate to insert "in fish as in other vertebrates" into the topic sentence "the present results demonstrate, however, that not all Purkinje cells respond identically to sensory or motor signals". The Discussion paragraph starting with "Evidence for different classes of Purkinje cells has also been found in the mammalian cerebellum" similarly ignores decades of physiological analyses of Purkinje cells that demonstrate differences in response properties. I suggest that the term "class" be defined more precisely and that the modular organization of inferior olivary projections to the cerebellum be explicitly discussed as it pertains to fish, birds, and mammals.

We think that some misunderstanding arose in the reviewer’s reading of the original text, as we were referring to electrophysiological activity, which are similar across Purkinje cells (i.e., they all make simple and complex spikes), but the reviewer may have been referring to differences in physiological responses (i.e. responses to different types of stimuli).

Nevertheless, the reviewer’s reaction also indicates an (actually uplifting) assumption that mammalian results generalize to the larval zebrafish, which took us by surprise, as we expected the assumption would be to the contrary. With this idea in mind, we have edited extensively throughout the manuscript to stress how the study is consistent with yet builds on existing mammalian and zebrafish literature; we have added 37 additional citations; and we have revised and expanded the text where previous studies of Purkinje cell diversity, including olivocerebellar modules, are mentioned. Although we acknowledge that our citation of the literature is still incomplete, we have tried to include a reasonable number of pertinent references. We have also consulted with three colleagues who are more familiar with cerebellar physiology than we are to try to ensure that we have covered the appropriate ground.

The meaning of the word “class” as used here—to denote a category or group with a common feature—is also stated explicitly in the present version.

12) There is a puzzling disconnect between the known anatomical projections of zebrafish Purkinje cells and the different classes of Purkinje cells reported in this paper. How do the different firing patterns evoked during visual stimuli, somatosensory stimuli, and fictive swimming relate to the differential projections of medial vs lateral Purkinje cells. A new paper from Sawtell's group describing activity of presumably the same Purkinje cells as those recorded for this study is relevant for this issue (https://www.ncbi.nlm.nih.gov/pubmed/27512018).

We are not quite sure what the reviewer is referring to. None of this work is in conflict with the paper from Nate Sawtell’s group (Scalise, Shimizu, Hibi and Sawtell, 2016). Purkinje cells innervate eurydendroid cells, whose projections were not studied in Scalise et al., or in the present manuscript. In our original manuscript, the section of the Discussion entitled “Topographical organization of the zebrafish cerebellum” discussed the present data in the context of work on projections out of the zebrafish cerebellum, which make a good case for a medial-lateral organization. We have added/revised text to clarify what is known about the projections of Purkinje cells and of eurydendroid cells.

13) The associative conditioning paradigm and resulting behavioral (nerve root activity) output are a bit puzzling and merit more discussion. The light is on for 2 seconds prior to the tail shock – this is a very long time for the cerebellum and seems particularly long in the short life of a baby zebrafish. What motivated this paradigm? Did a shorter visual stimulus (e.g. 500 ms or 1s) work as well? Does the apparently stochastic timing of the conditioned response reflect immaturity of the larval zebrafish cerebellum, or something about the prolonged timing of the visual stimulus? If they are conditioned responses, do they extinguish after repeated visual stimuli in the absence of somatosensory stimulation?

The 2-second light was used because of the slow processing of visual responses and because it gave good associative learning in pilot experiments along with an informatively long window in which we could observe Purkinje cell responses during the CS. Nevertheless, we recognize that 2 seconds seems long in comparison to mammalian studies of delay eyelid conditioning. We have done additional experiments and now include behavioral results with 1-sec and 0.5-sec light stimuli, reported in Figure 3 and the Results subsection “A cerebellar learning task in the zebrafish”. A 1-sec CS gave learned responses similar to 2-sec, while fewer fish learned and performed fewer learned responses with a 0.5-sec CS.

We also did experiments to test extinction, reported in the Results section. CRs were extinguished in all six fish that had learned to a criterion of making 3-5 consecutive CRs within 30 trials, and which were then tested with CS-only trials.

Please note, the timing of the CR was not stochastic, but the average latency decreased with training, as shown in Figure 3.

14) It would be very helpful to discuss what might be learned of general relevance for cerebellar function from studies of the larval zebrafish cerebellum in which firing rates are an order of magnitude lower than in mature cerebella from other vertebrates. Certainly this (and the recent Sawtell paper) raise intriguing questions about cerebellar evolution and whether there are fundamental computations that work robustly in both tiny sluggish cerebella and large speedy cerebella. How much do the conclusions depend on developmental stages? Please discuss the implications of multiply innervation by climbing fibers and low firing rates for the interesting and potentially general conclusions of this study about the dynamic changes in Purkinje cell spiking and functional role of simple spikes during associative conditioning.

The discussion has been edited to bring up more similarities and differences between zebrafish and mammals. Owing to space constraints and the already long manuscript, however, we did not delve too deeply into speculations about cerebellar evolution and fundamental computations.